# A potent broadly neutralizing human RSV antibody targets conserved site IV of the fusion glycoprotein

Aimin Tang[1,7], Zhifeng Chen[1,7], Kara S. Cox [1,7], Hua-Poo Su[2], Cheryl Callahan[1], Arthur Fridman[3], Lan Zhang[1], Sangita B. Patel[2], Pedro J. Cejas[1], Ryan Swoyer[1], Sinoeun Touch[1], Michael P. Citron[1], Dhanasekaran Govindarajan[1,5], Bin Luo[4], Michael Eddins[2], John C. Reid[2], Stephen M. Soisson[2], Jennifer Galli[1], Dai Wang[1], Zhiyun Wen[1], Gwendolyn J. Heidecker[1], Danilo R. Casimiro[1,6], Daniel J. DiStefano[1] & Kalpit A. Vora [1]

Respiratory syncytial virus (RSV) infection is the leading cause of hospitalization and infant mortality under six months of age worldwide; therefore, the prevention of RSV infection in all infants represents a significant unmet medical need. Here we report the isolation of a potent and broadly neutralizing RSV monoclonal antibody derived from a human memory B-cell. This antibody, RB1, is equipotent on RSV A and B subtypes, potently neutralizes a diverse panel of clinical isolates in vitro and demonstrates in vivo protection. It binds to a highly conserved epitope in antigenic site IV of the RSV fusion glycoprotein. RB1 is the parental antibody to MK-1654 which is currently in clinical development for the prevention of RSV infection in infants.

[1] Department of Infectious Diseases and Vaccines Research, Merck & Co., Inc, West Point, PA, USA. [2] Department of Structural Chemistry and Chemical Biotechnology, Merck & Co., Inc, West Point, PA, USA. [3] Department of Scientific Informatics, Merck & Co., Inc., Rahway, NJ, USA. [4] Department of Pharmacology, Merck & Co., Inc., West Point, PA, USA. [5] Present address: Janssen Research & Development, 1400 McKean Road, Ambler, PA 19002, USA. [6] Present address: Sanofi Pasteur, 1 Discovery Drive, Swiftwater, PA 18370, USA. [7] These authors contributed equally: Aimin Tang, Zhifeng Chen, Kara S. Cox Correspondence and requests for materials should be addressed to K.A.V. (email: kalpit.vora@merck.com)

Respiratory syncytial virus (RSV), a member of the *Pneumoviridae* family, is an enveloped virus with single-stranded non-segmented negative-sense RNA genome which has two subtypes, type A and type B, currently in circulation. The burden of RSV disease is the highest in the most vulnerable populations, particularly the very young, the elderly, and the immunocompromised. According to the CDC, ~177,000 adults over the age of 65 are hospitalized in the United States for RSV and 14,000 of those individuals die from the infection each year[1]. The morbidity in infants is also extensive; it is the most common cause of bronchiolitis, lower respiratory tract infections (LRTI), and hospitalization in infants in the first 6 months of life[2]. Studies have estimated that 74,000–126,000 infants are hospitalized in the US every year, which translates to an annual rate of 25–40 per 1000 infants[3]. Globally, RSV is estimated to cause 22% of all acute LRTI among children < 5 years old and 3.4 million episodes of severe acute LRTI[4]. Moreover, RSV infection is the leading cause of death in children less than 1 year worldwide[5]. Children infected with RSV also have a higher risk of subsequently developing chronic conditions including allergic rhino-conjunctivitis[6], recurrent wheezing, and asthma[7]. Taken together, these data demonstrate the tremendous burden of RSV especially on infants and the unmet medical need that this infection represents.

Despite over 50 years of global research efforts, there is no licensed vaccine for the prevention of RSV infection. An active vaccine against RSV would ideally be administered and be effective starting from birth to have the most significant impact on the pediatric disease burden; however, the immature immune system of infants along with high safety requisites at this very young age generally precludes this approach[8,9]. Recent research and development efforts have focused instead on either maternal vaccination, thereby boosting the levels of anti-RSV maternal antibody passed via placental transfer, or by passive immunoprophylaxis with an RSV neutralizing antibody administered directly to the infant after birth. To date, no RSV maternal vaccination clinical trials have met a primary efficacy endpoint, including the recent Phase III clinical study of ResVax (Prepare™ Trial, Novavax)[10]. Passive immunoprophylaxis with a humanized monoclonal antibody, palivizumab (SYNAGIS®, AstraZeneca) is available for high-risk infants for the prevention of serious lower respiratory tract disease caused by RSV infection. This approval provides proof of concept for passive immunoprophylaxis. However, clinical use of this antibody is limited to premature infants and other children at the highest risk[11]. Additionally; palivizumab has demonstrated limited clinical benefit and is not commercially feasible for all infants due to the once-a-month dosing requirements and cost[9,12,13]. Therefore, a more potent RSV antibody which can be administered to all newborn infants is needed, as it would provide nearly immediate protection to this vulnerable population.

The RSV fusion (F) glycoprotein is the current leading target for the majority of vaccines and immunotherapies under development. Based on natural immunity studies, the F protein is considered a key antigen for protective immunity[9]. The virus utilizes F protein to gain entry into cells and therefore contributes to viral spread within the host. RSV F and G proteins are the only two antigens which induce RSV-neutralizing antibody responses; however, the F fusion glycoprotein has a higher degree of sequence conservation among RSV strains (>90%)[14] and is more immunogenic and cross-protective compared to the G glycoprotein[9]. Importantly, the F protein of RSV has been validated as a target for the prevention of RSV disease in the clinic by the mAbs palivizumab and motavizumab[11,15].

The F protein exists as trimers in two forms, a metastable, prefusion form (Pre-F) and a highly stable post-fusion form (Post-F). The crystal structure of both of these forms has been solved[16–18] and the major antigenic sites that are exposed on each form have been designated by number[19]. Antigenic site I, II (the binding site of palivizumab), III, and IV are present in both the pre- and post-fusion F structures, whereas antigenic sites Ø and V (also defined as site VIII) are only exposed in the pre-fusion confirmation[16,18–20]. MEDI8897, an antibody under clinical development for the passive RSV immunoprophylaxis for infants, binds to antigenic site Ø[16,21]. Surveillance of RSV subtypes A and B clinical strains has clearly shown that the RSV virus can exhibit sequence polymorphisms which evolve over time, and that some antigenic sites exhibit more variability than others[22–24]. A study by A. Hause et al.[22], comparing the F protein sequences in over 1000 isolates found that site IV was highly conserved (>99%) across all genotypes. Furthermore, a study by V. Mas et al.[23] found that sites III and IV were the most conserved regions of the protein. Since a single amino acid change can have a high impact on antibody binding, a monoclonal antibody which binds to a highly conserved epitope is more desirable and decreases the risk of emerging antibody-resistant viruses.

Here, we describe the preclinical characteristics of RB1, a fully human IgG1 monoclonal antibody targeting a region of antigenic site IV of the RSV fusion protein. This antibody is ~50-fold more potent in vitro than palivizumab, is equipotent on both RSV A and B subtypes, has potent in vitro neutralization activities against a panel of diverse RSV clinical isolates, and provides in vivo protection against RSV infection in a cotton rat challenge model. In addition, we identify its binding epitope via several approaches including crystal structure analysis and demonstrate that RB1 binds to a highly conserved region in site IV of RSV F protein. RB1 is the parental antibody of the half-life extended version, MK-1654, currently in clinical development as a passive intramuscular immunization for the prevention of RSV infection in infants.

## Results

**Isolation of a potent RSV mAb from human memory B cells**. The human memory B-cell population is a rich source of naturally affinity matured and potent RSV neutralizing antibodies due to repeated RSV infection in adults. To clone a potent RSV neutralizing antibody from memory B-cells, we screened a cohort of adults for RSV neutralization titers, and isolated peripheral blood mononuclear cells (PBMC) from select donors. From these PBMCs, RSV-specific memory B-cells were sorted using a biotinylated trimer of RSV post-fusion glycoprotein as the antigen bait and cultured for 14 days for conversion to antibody-secreting cells. The resultant B-cell culture supernatants were screened for neutralization activity and binding to RSV fusion protein in ELISA. We then isolated and cloned the genes for associated paired heavy and light chains from neutralizing wells, and recombinantly expressed these fully human antibodies for further analysis (Supplementary Table 1). RB1 was selected for further characterization due to its potent in vitro neutralization activity. The antibody heavy and light chain variable region sequences are shown as aligned with the human antibody germline sequence (Supplementary Figure 1). The germline VH and VL gene belong to VH3-49*04 and KV1D-13*01 family respectively. H- and L-chain J regions belong to JH6*02 and JK5*01 respectively and the D segment was identified as D4-23*. The isotype of the antibody was IgG1.

**In vitro binding and neutralization activity of RB1**. We characterized the in vitro binding and neutralization activity of the purified recombinant antibody. RB1 equipotently neutralized laboratory RSV A (Long) and RSV B (Washington) strains with

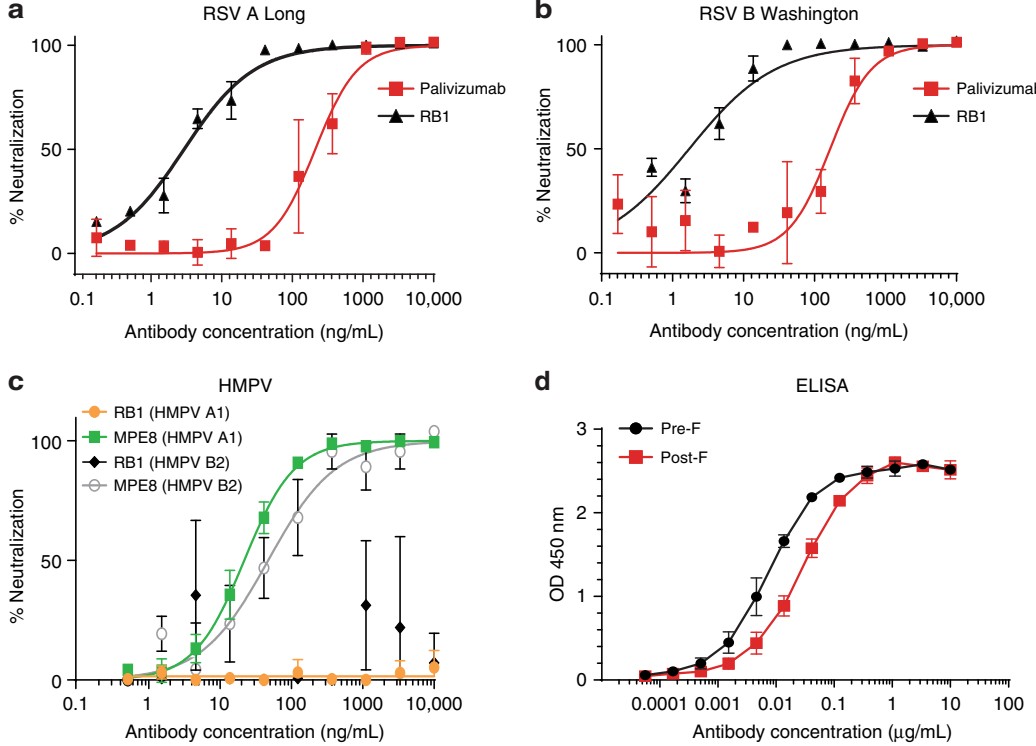

**Fig. 1** In vitro binding and neutralization. **a** The neutralization activity of RB1 against laboratory strains RSV A (Long) and **b** RSV B (Washington) with palivizumab as an assay control antibody. Assays were run in duplicate, showing error bars at mean with standard deviation (SD). An IC$_{50}$ was calculated using a log versus response variable slope 4 parameter fit curve and represents the concentration of antibody required for a 50% reduction in the RSV infectivity in a microneutralization assay. **c** The neutralization activity of RB1 against laboratory strains HMPV A and B with MPE8 control antibody. Assays were run in triplicate, showing error bars at mean with standard deviation (SD). An IC$_{50}$ was calculated using a log versus response variable slope 4 parameter fit curve and represents the concentration of antibody required for a 50% reduction in the HMPV infectivity in a microneutralization assay. **d** The enzyme-linked immunoassay (ELISA) binding curves of RB1 binding to the pre-F and post-F protein conformations. The assay was run in duplicate showing error bars at mean with SD. An EC$_{50}$ was calculated and represents the concentration of antibody required for a 50% reduction in binding. Source data provided as a source data file

| Table 1 SPR Affinity determination for RB1 binding to pre and post-F proteins | | | | | |
|---|---|---|---|---|---|
| | **Pre-fusion F Protein** | | | **Post-fusion F Protein** | | |
| | $k_{on}$ (M$^{-1}$s$^{-1}$) | $k_{off}$ (s$^{-1}$) | $K_D$ (pM) | $k_{on}$ (M$^{-1}$s$^{-1}$) | $k_{off}$ (s$^{-1}$) | $K_D$ (pM) |
| RB1 Fab | 7.35E + 06 | 1.64E−04 | 22 | 1.12E + 06 | 0.15 | 1.35E + 05 |

$K_D$ affinity or equilibrium dissociation constant, $k_{on}$ association rate, $k_{off}$ dissociation rate, M molar, pM picomolar, s seconds

an IC$_{50}$ of 2.9 ng/mL and 1.7 ng/mL, respectively (Fig. 1a, b), compared to palivizumab (IC$_{50}$ of 211.5 ng/mL RSV A and 166.3 ng/mL RSV B). This IC$_{50}$ is among the lowest of RSV neutralizing antibodies reported thus far, including those discovered from human RSV antibody repertoire analysis[25,26]. RB1 was then tested for neutralizing activity against a related virus, human metapneumo virus (HMPV), as some RSV antibodies exhibit activity against both, such as MPE8[27]. However, RB1 did not neutralize HMPV A or B strains (IC$_{50}$ > 10,000 ng/mL) as shown in neutralization assays and the control antibody MPE8 potently neutralized both strains (IC$_{50}$ 21.8 ng/mL and 42.1 ng/mL for strain A and B, respectively) (Fig. 1c). Binding affinities were measured using pre- and post-fusion conformation ELISAs and surface plasmon resonance (SPR). RB1 demonstrated strong pre- and post-fusion glycoprotein binding in the ELISA with a preference for the pre-fusion conformation (EC$_{50}$ of 7.4 ng/mL) as compared to the post-fusion conformation (27.1 ng/mL) (Fig. 1d). SPR analysis confirmed this preferential binding of RB1

to pre-fusion F protein, with a calculated $K_D$ of 22 pM, as compared to a $K_D$ of $1.35 \times 10^5$ pM to the post-fusion protein (Table 1, and Supplementary Fig. 2A, B). This difference in affinity is primarily accounted for by the slower off-rate of RB1 bound to the pre-fusion F glycoprotein ($1.64 \times 10^{-4}$ s$^{-1}$) relative to the post-fusion F glycoprotein (0.15 s$^{-1}$) (Table 1). The observed higher binding preference to post-fusion F in the ELISA compared to SPR could be explained due to avidity effect of full-length IgG used in ELISA as compared to Fab in the SPR analysis. The $K_{on}$ rates of RB1 Fab binding to preF and postF conformations are in the same order of magnitude ($7.3 \times 10^6$ vs $1.1 \times 10^6$ M$^{-1}$ s$^{-1}$), suggesting no significant steric hindrance in the postF conformation, as expected when binding a trimeric molecule. In contrast, the $K_{off}$ are quite different ($1.6 \times 10^{-4}$ vs 0.15 s$^{-1}$), indicating that the different dissociation rates are the main cause for the preF binding bias (Table 1).

RB1 was further evaluated for its activity and potency against a panel of 47 clinical isolates in an in vitro neutralization assay.

**Table 2 Data collection and refinement statistics (molecular replacement)**

|  | RB1 complex |
|---|---|
| *Data collection* | |
| Space group | P1 |
| Cell dimensions | |
| *a, b, c* (Å) | 118.2, 126.6, 148.2 |
| α, β, γ (°) | 87.2, 79.2, 86.3 |
| Resolution (Å) | 49.9–3.4 (3.5–3.4 Å) |
| $R_{sym}$ or $R_{merge}$ | 0.122 (0.523) |
| $I / \sigma I$ | 3.84 (1.55) |
| Completeness (%) | 98.92 (98.46) |
| Redundancy | 2.0 (2.0) |
| *Refinement* | |
| Resolution (Å) | 49.9–3.4 |
| No. reflections | 114445 |
| $R_{work}/R_{free}$ | 0.2453/0.2746 |
| No. atoms | |
| Protein | 41241 |
| Ligand/ion | 356 |
| Water | 0 |
| *B-factors* | |
| Protein | 80.86 |
| Ligand/ion | 93.01 |
| Water | – |
| R.m.s. deviations | |
| Bond lengths (Å) | 0.004 |
| Bond angles (°) | 0.74 |

Data were collected using a single crystal
Values in parentheses are for highest-resolution shell

This panel consisted of a broad range of clinical and laboratory RSV isolates from different years and locations and included different subtypes (24 RSV A and 23 RSV B). Additionally, each virus contained at least one amino acid change within the F fusion glycoprotein as compared to a laboratory reference strain. The F extracellular domain sequences from these clinical isolates were compared to the RSV F sequences reported in GenBank. The F extracellular domain sequences were clustered and visualized as a dendrogram, with the tested clinical isolate sequences displayed as the spokes on the outside of the circle (Fig. 2a). The sequences of the clinical isolates were distributed throughout the phylogenetic tree of the RSV sequences, demonstrating that the viruses used in this analysis represent a wide breadth of RSV isolate F sequences reported in the literature. RB1 potently neutralized each of the RSV A viruses with a calculated median $IC_{50}$ of 3.71 ng/mL (range 0.46–11.11 ng/mL) and the RSV B viruses with a calculated median $IC_{50}$ of 4.46 ng/mL (range 0.58–29.65 ng/mL) (Fig. 2b). This data confirms that RB1 is equipotent on RSV subtypes A and B and shows that the antibody has potent neutralization activity against a broad panel of RSV viral isolates.

**Determination of the binding site by shotgun mutagenesis.** The in vitro binding data demonstrated that RB1 binds to an area of the RSV F glycoprotein that is accessible in both the pre- and post-fusion conformations. To further map the binding to a specific region or antigenic site of the RSV F glycoprotein, an analysis of contact residues was performed using an alanine scanning shotgun mutagenesis methodology[28]. Three hundred sixty-eight (368) surface-exposed residues were selected based on the crystal structures of the pre-fusion and post-fusion RSV F glycoproteins[16,18] and expression constructs were generated to create a comprehensive mutation library where every residue of

interest was individually mutated to an alanine, and an alanine to a serine. Library screening was performed with RB1, and two residues of the RSV F glycoprotein, arginine 429 and isoleucine 432, were identified as critical residues and mutations of either one resulted in a loss of RB1 binding (Supplementary Fig. 3). This epitope on the RSV F glycoprotein overlaps with the region defined as site IV[26,29,30].

**Crystal structure analysis of RB1 binding to RSV F.** The crystal structure of the RB1 fragment antigen-binding (Fab) region bound to the RSV F glycoprotein was obtained to visualize the antigen-antibody interaction at atomic resolution and further characterize the amino acids that make up the binding epitope. Crystal structure analysis revealed that three copies of RB1 were bound to the trimer of the stabilized pre-fusion RSV F glycoprotein (DS-Cav1) (Fig. 3a, Table 2). Both the heavy and light chains of the antibody, primarily through the complementarity-determining region (CDR) loops, interact with the protomer 1 subunit of the F glycoprotein at site IV (Fig. 3a). A stereo image of the electron density for RB1 binding to pre-fusion F protein is shown in Supplementary Fig. 4.

The epitope is largely composed of interactions within the protomer 1 subunit of RSV F glycoprotein, specifically the residues asparagine 426, lysine 427, asparagine 428, arginine 429, isoleucine 432, lysine 433, aspartic acid 440, tyrosine 441, serine 443, lysine 445, glycine 446, and valine 447. From the neighboring protomer in the fusion protein, the residues glutamic acid 161 and serine 182 are within interaction distance of RB-1 based on a 3.5-angstrom distance cutoff (Fig. 3b).

On RB1, the paratope is composed of several residues from both heavy and light chains. From the heavy chain, serine 28 and aspartic acid 30 of the CDR1 loop, tyrosine 56 of the CDR2 loop, and glycine 105 and 106 asparagine 107, serine 108, and tyrosine 110 of the CDR3 loop comprise the paratope. From the light chain, arginine 30 of the CDR1 loop, aspartic acid 50, glutamic acid 55, and tyrosine 56 of the CDR2 loop, phenylalanine 91 and leucine 92 of the CDR3 loop form the paratope. Additionally, there are non-CDR residues that can interact with the RSV F protein based on a 3.5-angstrom distance cutoff, including lysine 39, glycine 57, serine 63, and 65 (Fig. 3b).

The crystal structure of another site IV antibody, 101F, was previously solved as the antibody bound to a peptide from the RSV-F protein[30]. Based on alignment of this peptide, a comparison of the full epitopes of 101F and RB1 demonstrate overlapping epitopes with interactions unique to each antibody (Fig. 3c). The lower binding affinity to post-fusion F relative to pre-fusion F, as determined by SPR, can be rationalized by anticipated changes to the structure upon transition between the forms. The post-fusion F structure (pdb code:3rrr)[18] was superposed on the prefusion F structure based on residues 425–447, which encompass 12 of the 14 interacting residues on F. In Fig. 3d, the structure with pre-fusion F is shown on the left with in ribbon with RB1 in cartoon (cyan and purple). The interacting residues from the neighboring monomer of F, Glu 161, and Ser 182, are shown in sticks in gray. On the right side of the figure is the superposed post-fusion F protein but depicting that the neighboring monomer is no longer near the antibody. Additionally, a strand (shown in cartoon depiction) spanning residues 464–470 is positioned within 4 angstroms of the antibody. In the post-fusion conformation, the strand swings up towards the fusion bundle. Figure 3e shows the overlay from a farther perspective. Residues that form the epitope are depicted as spheres. On the right, the shift in the neighboring monomer and residues 161 and 182 now form the fusion bundle. Of the 1006 Å$^2$ interface made between the antibody and the antigen, 191 Å$^2$ is

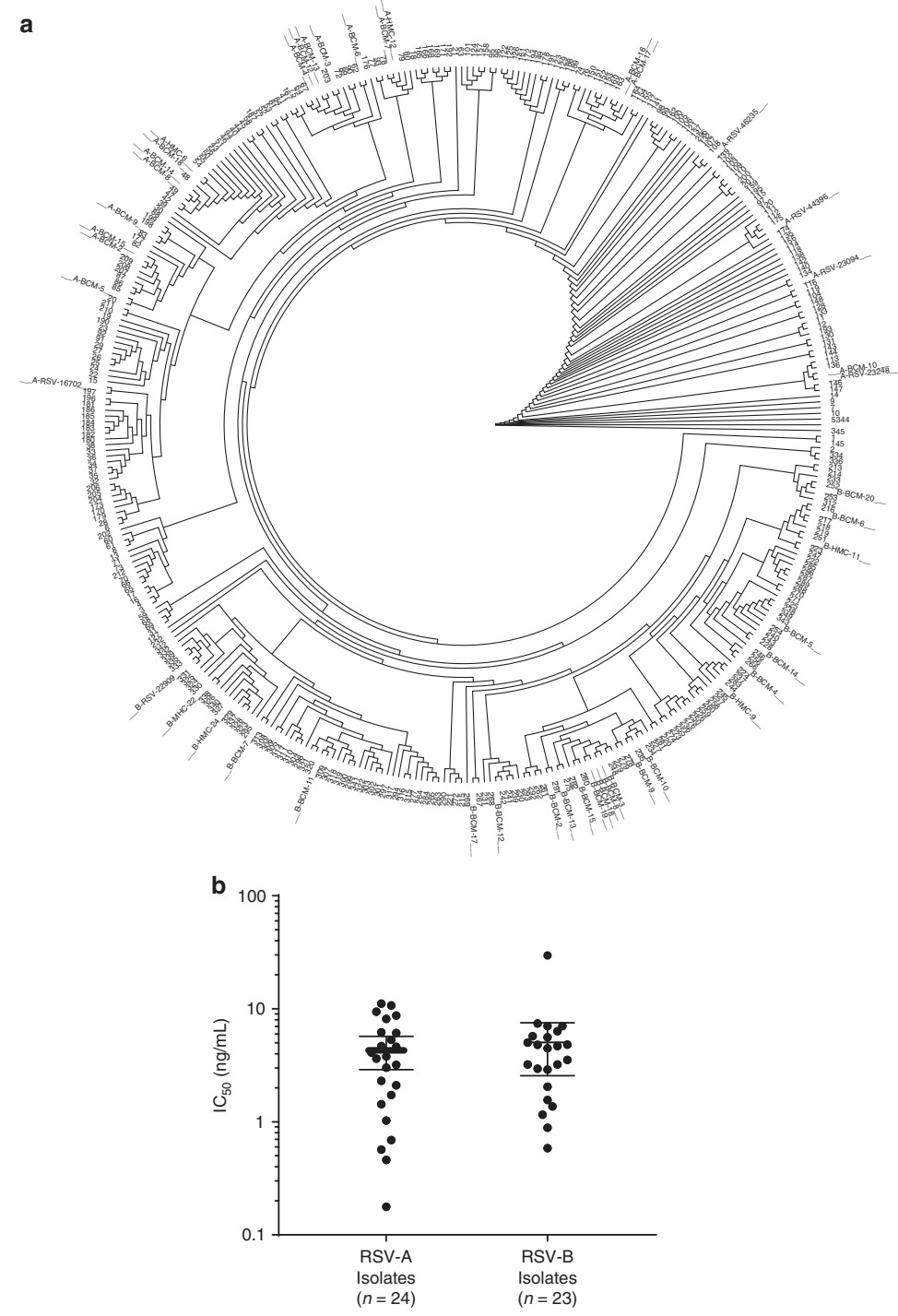

**Fig. 2** Neutralization of a diverse set of clinical isolates. **a** The clinical isolates used to evaluate RB1 neutralization activity were displayed as a dendrogram to evaluate sequence diversity. A phylogenetic tree of 345 unique GenBank sequences is represented in the inner part of the circle while the fusion protein sequences of 46/47 RSV A and B clinical isolates are marked as spokes on the outside of the circle (1 clinical isolate contained an incomplete sequence). Source data (clinical isolate sequences and accession numbers) are provided as a source data file. **b** RB1 was assessed for its ability to neutralize RSV clinical isolates using an in vitro microneutralization assay. A panel of 47 RSV clinical isolates containing amino acid changes in the F fusion glycoprotein were tested in the in vitro neutralization assay. $IC_{50}$ was calculated and represents the concentration of antibody required for a 50% reduction in the RSV infectivity. The error bars represent the geometric mean with 95% confidence intervals

made to the neighboring monomer, which is no longer available in the post-fusion form[31].

**Conservation of RSV F protein site IV and the RB1 epitope**. It is important to understand the prevalence of naturally occurring amino acid (a.a.) substitutions in reported RSV strains to evaluate

the frequency of reported polymorphisms in the RB1 binding epitope and to assess the potential risk for the emergence of RB1-resistant viruses. Therefore, we performed an analysis of 3600 complete RSV F glycoprotein sequences obtained from GenBank to evaluate the reported polymorphisms. Out of these, 3058 contained the complete extracellular domain and had no

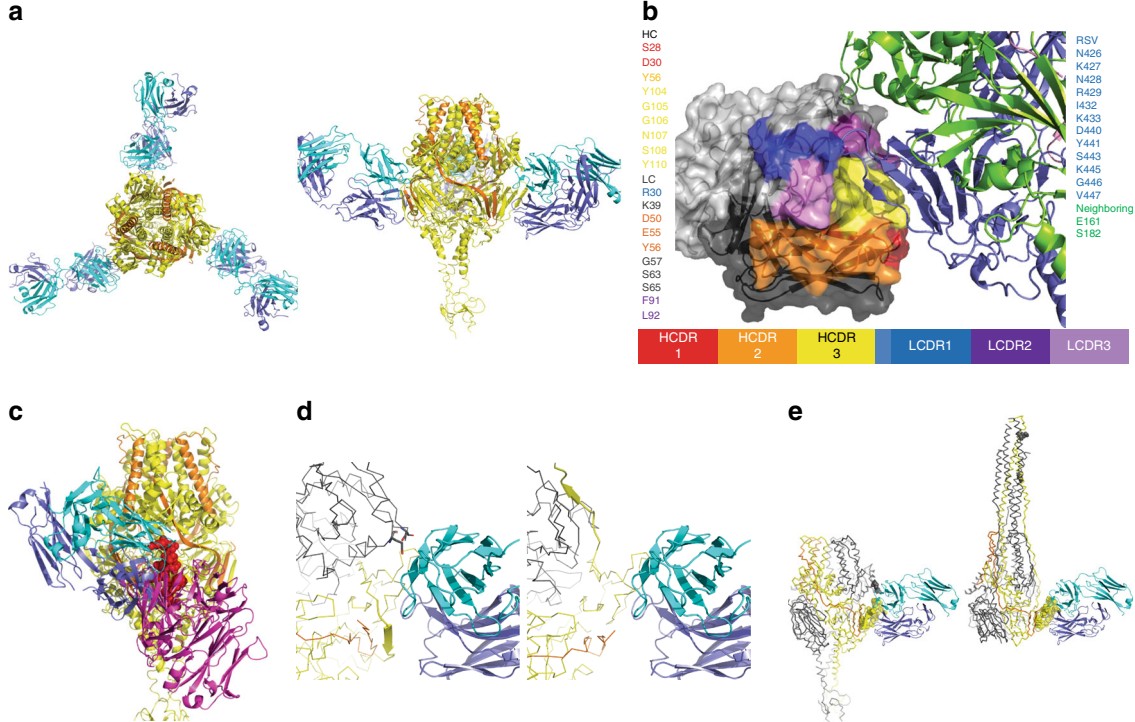

**Fig. 3** Crystal structure of the RB1 Fab and RSV Pre-fusion F complex. **a** Interaction of RB1 with the RSV pre-F trimer. Structure of RSV pre- F trimer (protomer 1: yellow, protomer 2: orange) bound to three copies of RB1 (heavy chain: blue, light chain: cyan) as viewed down the 3-fold axis (left) and rotated 90º to view from the side (right). **b** Close-look of the RB1-RSV pre-F interaction. Interaction of RB1 and RSV-F: RSV F protein is shown in ribbons on the right with the protomer1 subunit in blue and the protomer2 in green. RB1 is shown as surface representation with the CDRs colors (HC: CDR1: red, CDR2: orange, CDR3: yellow, LC: CDR1: blue, CDR2: dark purple, CDR3: light purple). **c** Comparison of RB1 and 101F binding. A side view of the RB1 RSV-F interaction with the 101 F interaction modeled in overlay. The RSV-F trimer: RB1 interaction is shown as colored previously (protomer 1: yellow, protomer2: orange, heavy chain: blue, light chain: cyan). The 101F structure was superimposed based on the peptide derived from residues, 427–436 of the F protein (shown as spheres in red). The resulting pose of the 101F Fab is shown in purple. **d** Close up view of the interaction interface between prefusion F and RB1 (left) and the superposed post-fusion F and RB1 (right). RB1 is depicted in cartoons and held fixed in both images with heavy chain in purple and light chain in cyan. The RSV-F is shown in CA ribbon representation with one strand spanning residues 464–470 depicted in cartoon representation to illustrate the shift between prefusion and post-fusion conformations near the epitope. The monomer that mediates the main interaction is colored in yellow and the other two monomers of the trimer are shown in gray. Two residues of the neighboring monomer, Glu 161 and Ser 182, are shown as sticks and no longer near the antibody in the post-fusion conformation. **e** A zoomed out view of the prefusion F: RB1 complex is shown on the left, in contrast to a superposed post-fusion F with RB1 on the right. Residues of F within 3.5 angstroms of the antibody in the prefusion form are shown as spheres

**Table 3 Conservation of RSV F protein amino acids in the RB1 binding epitope**

| Count | Frequency | Amino acid position 426 427 428 429 /432 433 / 440 441 / 443 / 445 446 447 | Subtype | Country of Isolation | Year | Accession # |
|---|---|---|---|---|---|---|
| 3054 | 99.86% | NKNR/IK/DY/S/KGV | RSV A, B | | | AHY2137[a] |
| 1 | 0.03% | NKNR/IK/DY/S/K*E*V | RSV A | USA | 1982 | AHY21320 |
| 1[b] | 0.03% | NKNR/IK/DY/S/KG*M* | RSV A | n/a | n/a | AAB59858 |
| 1 | 0.03% | NKNR/IK/DY/S/*R*GV | RSV A | USA | 1987 | AMA67163 |
| 1 | 0.03% | NKNR/IK/*N*Y/S/KGV | RSV B | China | 2016 | AVQ93607 |

Italics means a change in the amino acid from the reference sequence AHY2137
/ = non-continuous amino acid sequence. Table heading indicates amino acid positions depicted
[a]A representative accession number for this sequence
[b]Laboratory strain A2

ambiguous amino acids in antigenic site IV. Comparison of these sequences in the binding area of RB1, as determined by crystallographic analysis, demonstrated that amino acids 426–447 in site IV of RSV fusion protein is conserved in 3054 of 3058 sequences with 99.9% identity (Table 3). Each of the four variants were only reported once suggesting that they exist at very low frequencies or are potentially a result of sequencing error. Overall, this analysis shows that the binding site of RB1 is highly conserved and is,

therefore, a desirable target for prophylactic immunotherapy for RSV.

We further analyzed the GenBank sequencing data for the entire RSV fusion glycoprotein sequence (a.a. 1 through 574) to compare the variation in site IV to all other areas of the protein, including antigenic site Ø and site II, the binding site for palivizumab. The frequency of each of the amino acid (s) at each position is depicted as a logo plot (Supplementary Fig. 5).

**Table 4 Monoclonal antibody-resistant mutants (MARMS) for RSV A and B**

| Designation | In vitro selection virus | RSV F Sequence Change | Neutralization IC$_{50}$ (ng/ml) |
|---|---|---|---|
| RB1-A1 | RSV Strain A2 | G446E | >1000 |
| RB1-A2 | RSV Strain A2 | S443P; K445N | >1000 |
| RB1-A5 | RSV Strain A2 | S443P; G446 | >1000 |
| RB1-B6 | RSV Strain A2 | S443P | >1000 |
| RB1-P1A1 | RSV B Washington | S443P | ND |

ND not tested

Assessment of these reported sequences shows that site IV has a higher degree of sequence homology as compared to site Ø or site II.

**In vitro selection and characterization of RSV MARMS to RB1.** In order to identify critical residues which are susceptible to escape of RB1 neutralization, we generated RSV monoclonal antibody-resistant mutants (MARMs) for RSV A and B subtypes. A total of four MARMs were identified for RB1 after extensive in vitro selective pressure with an RSV A2 laboratory strain. These four MARMs were sequenced and found to have mutations located in the binding epitope region. Two MARMS contained a single a.a. mutation; MARM RB1-A1 with a mutation in position 446 from glycine to glutamic acid (G446E), and MARM RB1-B6 with a mutation in position 443 from serine to proline (S443P) (Table 4). MARM RB1-A5 included both of those mutations, G446E and S443P. Lastly, MARM RB1-A2 contained the S443P mutation as well as a change in position 445 from lysine to asparagine (K445N) (Table 4). An in vitro assay confirmed that the viruses were resistant to neutralization by RB1 (IC$_{50}$ > 1000 ng/mL) (Table 4) as well as binding as measured by an enzyme-linked immunoassay (Supplementary Fig. 6).

Each MARM virus was assessed for viral growth characteristics in HEp-2 cells. MARM RB1-A1, RB1-A2, and RB1-A5 viruses had slower in vitro growth rates as well as lower peak titers as compared to the RSV A2 Long laboratory virus, while MARM RB1-B6 demonstrated a similar growth pattern, as compared to the RSV A2 laboratory strain (Supplementary Fig. 7). The data for the RB1-A1, RB1-A2, and RB1-A5 MARM viruses suggest reduced viral fitness compared to the RSV A2 laboratory virus in vitro.

The in vitro selection experiments were next performed using the RSV B Washington laboratory strain to evaluate the susceptibly of escape mutations from the RSV B subtype. A single MARM, with the mutation in position 443 from serine to proline (S443P) was identified (Table 4).

In order to monitor for the prevalence of RB1 resistance-associated mutations in naturally occurring sequences, we compared the amino acid changes identified in MARMS to our bioinformatic analysis of the GenBank sequences. There were no reported polymorphisms at the site of our most frequently identified MARM at position 443 from serine to proline (S443P), or any report of a change at position 445 from lysine to asparagine (K445N). A single report of a mutation in position 446 from glycine to glutamic acid (G446E), was reported in 1982 (accession # AHY21320).

**Prophylactic efficacy in the cotton rat challenge model.** We next evaluated the prophylactic antiviral activity of RB1 in the cotton rat (*Sigmodon hispidus*) model of RSV infection. This animal challenge model has shown good translatability in the

projection of target therapeutic pharmacokinetics and efficacy in humans for other anti-RSV antibodies[32,33]. In this study, weight-based doses of RB1 were administered as a single intramuscular injection of antibody at day 0. The following day serum was collected for determination of circulating concentrations of RB1. Immediately after blood sample collections, each animal was sedated and challenged, intra-nasally, with $1 \times 10^5$ pfu of either RSV strain A2 or strain B 18537. Four days post viral challenge, the animals were euthanized, and nose and lung tissue collected to assess antibody efficacy by measuring RSV infectious titers in these tissues. The RB1 antibody demonstrated lower respiratory tract (lung) EC$_{50}$ values of 1.1 μg/mL and 1.9 μg/mL for the RSV A and RSV B strains, respectively (Fig. 4a, c). RB1 administration resulted in upper respiratory tract (nose) EC$_{50}$ values of 9.9 μg/mL and 8.5 μg/mL for the RSV A and RSV B strains, respectively (Fig. 4b, d). These in vivo data demonstrate that RB1 exhibited potent and dose-dependent antiviral activity in the lungs and nose after infection with RSV subtypes A and B with generally equal potency between the subtypes.

**Evaluation of the role of effector functions.** To evaluate the role of the effector function in RB1 in vivo protection and confirm the mechanism of action, the mAb was modified with amino acid substitutions in the fragment crystallization (Fc) region, where leucine 234 was modified to alanine and leucine 235 was modified to alanine, e.g., L234A and L235A (LALA). These mutations have been shown to both greatly reduce the binding to Fc gamma receptors (FcγR) and to attenuate complement dependent cyto-toxicity (CDC) and FcR-mediated cell cytotoxicity (ADCC) thereby reducing the effector function of the antibody[34,35]. We then performed an additional in vivo study in the cotton rat challenge model with strain A2 to directly compare the RB1-LALA mAb with RB1. Animals treated with LALA-modified RB1 mAb demonstrated similar lung protection which was not significantly different than -cotton rats treated with the unmodified antibody (1.3-fold difference in EC$_{50}$, $p = 0.758$, two-tailed unpaired $t$-test) (Fig. 4e). As a control, we immunized another set of animals with palivizumab or the LALA-modified version. In contrast to RB1, the lung EC$_{50}$ of palivizumab was significantly different (5-fold higher, $p = 0.0028$, two-tailed unpaired $t$-test) in the animals dosed with the LALA mutated palivizumab as compared to the unmutated version (Fig. 4f). These data suggest that effector function is not required for RB1 in vivo efficacy in the cotton rat challenge model and the efficacy is originating from the antibody variable region likely by interfering with the function of the F protein-mediated fusion process.

**Discussion**
Advances in immunological techniques in the last decade have enabled us to harness the natural immune repertoire after infection as a source of highly potent antibodies from human memory B-cells. Particularly for RSV, which causes repeated infections throughout life, the memory B cell pools is a rich source of affinity matured and potent neutralizing antibodies. Here we describe RB1, an anti-RSV F protein antibody derived from an adult memory B-cell without any additional in vitro sequence optimization. The antibody binds to an epitope within antigenic site IV, previously referred to as site C. Site IV spans the residues from 422 to 468 and is the target of several well-characterized antibodies including MAb19, 101F, and 3M3[9,26,36]. MAb19 was humanized and tested in clinical trials but failed to show efficacy which was attributed to a lack of potency[37].

RB1 was isolated with post-F protein, although preferential binding is demonstrated to the pre-fusion conformation (Fig. 1d). This strong binding and affinity likely contribute to the potent

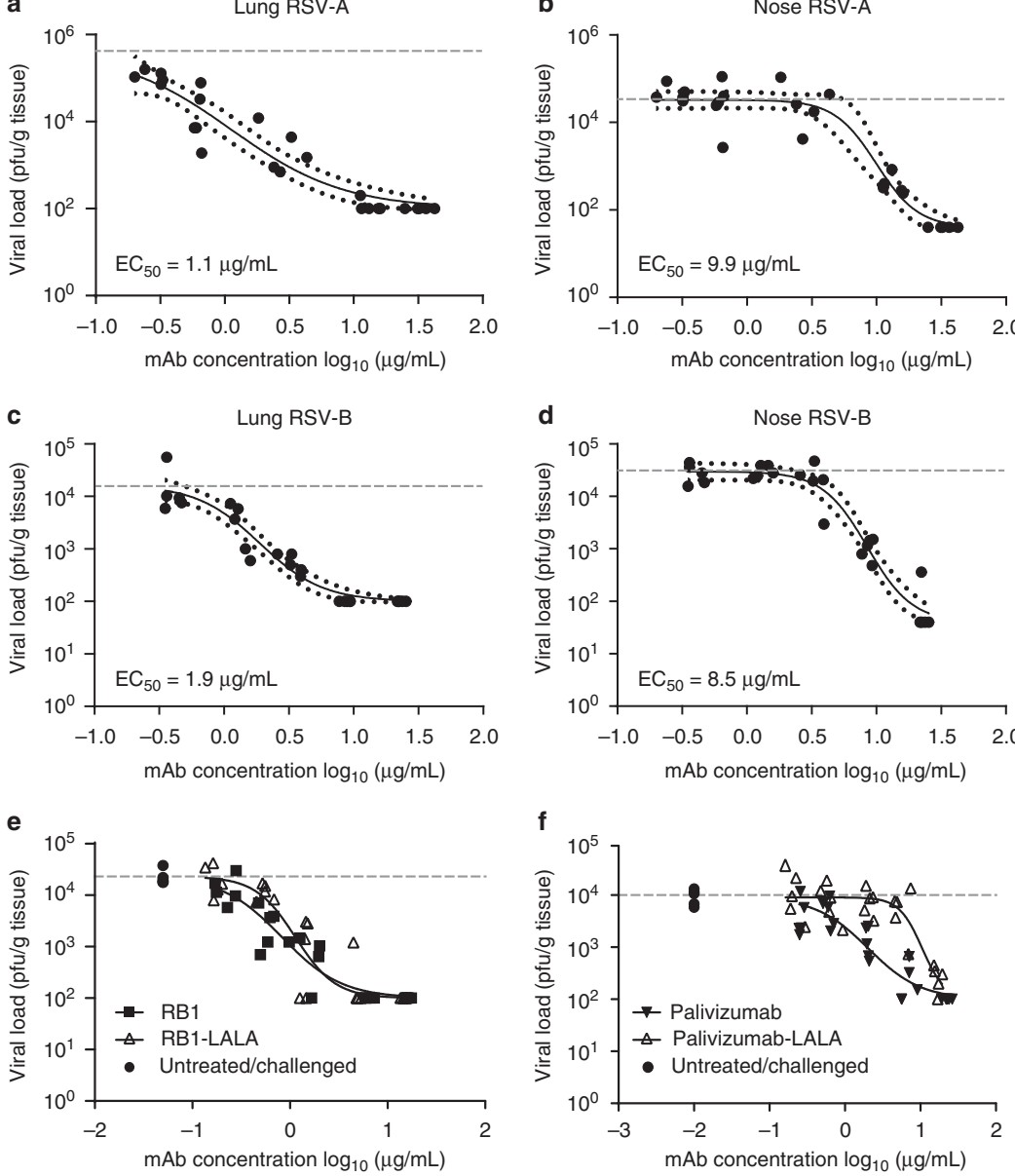

**Fig. 4** Efficacy of RB1 in the upper and lower respiratory tracts of cotton rats. Cotton rats were administered RB1 by intramuscular injection with a dose titration and blood was collected for evaluation of serum antibody concentrations the following day. Immediately after blood sample collections, each animal was challenged intra-nasally with $1 \times 10^5$ plaque-forming units (pfu) of RSV A2 or B 18537 strains. Four days post-challenge, animals were euthanized, blood collected for evaluation of antibody concentrations, and nose and lung tissue collected to assess mAb efficacy by measuring RSV infectious titers in both the nasal and pulmonary tissues. Depicted are the log pfu per gram of tissue for each strain for lung and nose tissues against the log of day 1 serum concentrations in µg/mL. Points represent observed data (5 animals per dose and 5 untreated per experiment) and lines represent non-linear regression fit of the inhibitory sigmoidal dose-response analysis with variable slope depicted with 95% confidence bands for RSV A lung (**a**), RSV A Nose (**b**), RSV B lung (**c**), and RSV B nose (**d**). **e** Shows similar analysis after dosing with RB1 or RB1-LALA, which contains mutations to decrease Fc function activity of antibodies, and an analysis of palivizumab as compared to palivizumab-LALA (**f**). The horizontal dotted lines in all graphs represent the average log pfu/g tissue for the naïve challenged controls in that experiment

neutralizing characteristics of the antibody and suggests that the antibody was probably elicited, or affinity matured by pre-F on the virus particle in vivo. Here we showed that RB1 was able to potently neutralize a diverse panel of clinical isolates with calculated median $IC_{50}$ of 3.7 ng/mL for RSV A and 4.4 ng/mL for RSV B. This is among the better potencies of reported RSV F antibodies. For example, in a large-scale RSV antibody repertoire analysis from adult B-cells reported by M. Gilman et al. fewer than 10 out of 364 antibodies had potencies of 6 ng/mL or less on RSV A or B[25]. That study also showed that most of the potent

antibodies (< 50 ng/mL) were pre-F specific, targeting site Ø and V. In the past few years, the general consensus in the RSV field was that site Ø and V were the targeting sites of the most potent neutralizing antibodies;[19] however, more recently there has been a realization that although rarer in frequency, highly potent site IV human antibodies also exist[26]. Jarrod Mousa et al. recently reported such a mAb with the discovery of 3M3[26].

In addition to the in vitro potency, we also evaluated the in vivo protection in the cotton rat prophylactic challenge model. This demonstrates that RB1 confers potent lung protection, as

well as inhibition of viral replication in the upper airway, a feature which is lacking with palivizumab[21]. Furthermore, we evaluated the role of Fc function by comparing RB1 prophylactic protection compared to a LALA mutated antibody and found that antibody effector function was not required for protection. The role of Fcγ receptor-mediated immunity in RSV is still being explored; however, some data suggest that it could contribute to disease pathogenesis[38].

The crystal structure of RB1 Fab in complex with DS-Cav-1 shows the binding at antigenic site IV contacting two adjacent fusion protomers. The antibody light chain engages an adjacent protomer; however, its role in determining preferential binding to pre-F as compared to post-F is not yet clear. This RB1 interaction with the fusion protein was different than the epitope of a well-characterized site IV antibody, 101 F. RB1 has a much more potent neutralization activity against RSV (IC$_{50}$ of 2.9 ng/mL for RSV A) as compared to 101F (IC$_{50}$ of ~150 ng/mL for RSV A Long)[26], which is likely a result of these differences in binding interactions between the two antibodies and the fusion glyco-protein. Furthermore, 101F has activity against human metap-neumovirus (HMPV), while RB1 does not.

The binding region of a potential immunoprophylaxis for RSV is critically important since epidemiological studies have clearly shown the evolution of sequence drift over time; thereby raising the concern for the emergence of antibody-resistant mutations[22–24]. A site V antibody, Suptuvumab (REGN2222), was recently discontinued after failing to meet the primary end-point in a phase III clinical trial, further highlighting the importance of resistant variants in the RSV field[39].

Here, we show a frequency evaluation of reported amino acid polymorphisms from over 3,000 sequences in GenBank. The binding region of amino acids in the range of position 426–447 were conserved with 99.9% identity (Table 3). Sequence analysis across the entire RSV F sequence shows that amino acid positions at site IV are more conserved than at sites Ø, II, and V (Sup-plementary Fig. 5). A few other recent publications reporting on sequence variability across the RSV fusion protein contain similar conclusions. A study by A. Hause et al.[22] compared the F sequences of over 1000 viral isolates found that site IV was highly conserved (>99%) across all genotypes. Furthermore, a study by V. Mas et al.[23] found that sites III and IV were the most con-served regions of the protein. They further argued that pre- and post-F share secondary structural elements within sites III and IV and these constitute the inter-protomeric cavity of the protein. Moreover, the parallel strand of the C-terminus F1 and the short alpha helix unravel to form the 6-HB motif at site IV. Thus, these sites may be less tolerant to changes in sequence, as such changes may render the virus unable to transition between the pre- and post-F conformations. Sites III and IV of the protein are con-served to some extent between RSV A, B, and the related virus HMPV; therefore, they may also be required for genetic stability. In contrast, placement of site Ø and V on the apex region may require structure flexibility resulting in less conserved sequences[23].

A recent study of circulating RSV strains in the United States from 2015 to 2017 conducted by B. Lu et al.[24] showed some variation in site IV in the RSV B strains, but the amino acid positions of these variants ranged from positions 462–467 and are, therefore, well outside of the RB1 binding region. Overall, the binding epitope for RB1 in antigenic site IV is conserved; how-ever, as for any monoclonal antibody continued strain surveil-lance efforts will be important to monitor for RSV evolution.

We also report the generation of four RSV monoclonal anti-body resistant mutants (MARMS) for RB1 and their growth kinetics in vitro. This data may provide us with additional gui-dance for RSV strain surveillance and identify critical binding

residues, which are more biologically relevant as compared to the shotgun mutagenesis approach where the mutations are artifi-cially created and restricted to alanine. Importantly, MARMS A1, A2, and A5 demonstrated reduced viral fitness to the parental antibody, and B6 replicated at about the same growth rate in vitro. This data suggests that the MARMS are unlikely to exhibit increased growth kinetics advantage over naturally occurring strains in vivo. The two amino acids R429 and I432 found from the shotgun mutagenesis were not identified in our in vitro selection pressure studies, nor were the K433T or R429S MARMS reported for 101F and RSV 19, respectively, suggesting differences in binding epitopes in addition to potency differences between these site IV antibodies and RB1[36,40].

In summary, this report describes the isolation and pre-clinical properties of a potent and broadly neutralizing fully human antibody targeting conserved antigenic site IV of the RSV F protein. The antibody binds to Pre-F with higher affinity than Post-F predominantly due to slower off rates for the former. The antibody demonstrated potent in vivo protection in the cotton rat model and does not require effector functions for its anti-viral activity. In total, these properties warrant its clinical testing for the prevention of RSV infection in at risk populations, including infants. To this end, a version of RB1, MK-1654, which contains Fc mutations to enhance its half-life, is currently in development as a passive intramuscular immunoprophylaxis for the prevention of RSV infection in infants.

## Methods

**Human subjects.** Informed consent for volunteers was obtained in accordance with the Helsinki Declaration of 1975 (approved by the Institutional Review Board of Merck Sharp & Dohme, a subsidiary of Merck & Co., Inc., Kenilworth, NJ, USA). PBMC from select donors were purified from blood collected in EDTA tubes by density gradient centrifugation in histopaque over Accuspin$^{TM}$ tubes (Sigma Aldrich, Cat. No: A2055) according to the manufacturer's instructions. PBMC were then frozen in 90% heat inactivated FBS supplemented with 10% dimethyl sulf-oxide and stored in liquid nitrogen until thawed for use in experiments.

**RSV post-Fusion F specific memory B cell isolation.** The method for isolating the post-F specific memory B cells and antibody from them was done similarly to the method reported in Cox et al.[41]. Specifically, cryopreserved PBMC were thawed on the day of sorting, washed with sterile PBS supplemented with 1% FBS, and incubated for 25 min with biotinylated Post-F protein. Cells were washed and stained with anti-CD3 BV421 (cat. #562426, BD Biosciences), anti-CD19 FITC (cat. #555412, BD Biosciences), anti-IgG APC (Cat. #550931, BD Biosciences) and streptavidin-PE (cat #349023, BD Biosciences) for 25 min each at the manu-facturer's recommended volume per test and washed. CD3−/CD19 + /IgG+/Post-F protein-binding cells were sorted with a BD FACS Jazz in single-cell mode into a 96-well plate. The sorted cells were cultured in RPMI with FBS, in the presence of irradiated CD40L-expressing feeder cells at a concentration of $4.0 \times 10^4$ cells/well (in-house made cell line) and IL-21 (Sino Biologicals cat. #10584-HNAE-20) at 50 ng/mL for conversion to antibody-secreting cells. Supernatants were transferred into new plates and assayed for RSV F protein binding and virus neutralization activities. Cell pellets in plates were lysed in 50 μl per well of RLT buffer (Qiagen) supplemented with 1% 2-mercaptoethanol (2-ME) (Sigma Aldrich Chemicals) and quickly frozen on dry ice. Plates were then transferred into a −80 °C freezer for storage.

**Expression and purification of RSV fusion proteins.** Plasmids encoding mam-malian codon-optimized RSV F pre-Fusion (DS-Cav1) and post-Fusion (FΔFP) proteins were used to transfect Expi 293 F cells (ThermoFisher cat. #A14527), and proteins were purified from culture supernatants[18,17]. Specifically, cell culture supernatants were harvested day 3 (FΔFP) or 7 (DS-Cav1) post-plasmid trans-fection, and RSV F proteins were purified using Ni-Sepharose chromatography (GE healthcare). Frozen cell culture supernatants were thawed and re-clarified at 14,260×g for 30 min at 5 °C. Supernatant was pH adjusted with addition of 1 M HEPES, pH 7.5 (typically 50 mL/L), and 2 M Imidazole (pH 7.5) was added to a final concentration of 10 mM. Post addition, the supernatant was stirred and fil-tered through a 1 L Nalgene Rapid-Flow (0.45 micron / PES) vacuum filter unit. RSV F proteins were purified using Ni-Sepharose chromatography (GE healthcare). FΔFP was further purified by Strep-Tactin chromatography (Strep-Tactin Super-flow Plus, Qiagen). Tags were removed from DS-Cav1 and FΔFP by overnight digestion with thrombin. To remove IMAC contaminants and uncleaved F protein, DS-Cav1 was subjected to a second Ni-Sepharose chromatography step. Both DS-

Cav1 and FΔFP were purified by gel filtration chromatography (Superdex 200, GE Healthcare) and were stored in a buffer of 50 mM HEPES pH 7.5, 300 mM NaCl. All column purifications were performed on a GE Healthcare AKTA Purifier System at room temperature. All buffers were vacuum filtered before use (0.22 micron, CA).

**RSV Fusion protein ELISA immunoassay.** Nunc C96 Maxisorp Nunc-Immuno Plates were coated with 50 μl per well of RSV Pre-F or Post-F protein at 1 μg/ml at 4 °C overnight. Plates were washed in PBS/0.05% Tween 20 and then blocked in PBS/0.05% Tween 20/3% non-fat milk at room temperature for 1 h. 50 μl per well of 3-fold serially titrated mAbs were then added and incubated at room temperature for 90 min. Plates were washed and 50 μl per well of HRP-conjugated goat anti-human IgG (1:2000) (Southern Biotech cat. # 2040–05) was added. After 60 min incubation at room temperature, plates were washed and 100 μl per well of SuperBlu-Turbo TMB Solution (ViroLabs) was added. Plates were incubated at room temperature for 5 min and the reaction was stopped by adding 100 μl per well of Stop Solution for TMB ELISA (ViroLabs). Plates were then read on VICTOR Multilabel Counter (Wallac/Perkin Elmer) at 450 nm.

**Recovery of antibody sequences from memory B cell culture.** The frozen-stored single-cell sorted and cultured 96 well plates containing lysed B cells in RLT buffer were thawed at room temperature. Total RNA extraction for immunoassay hits was performed with Qiagen RNeasy Micro Kit (Qiagen) following manufacturer's instructions.

The extracted total RNA was used as the template in RT-PCR to amplify human antibody genes with Qiagen One-step RT-PCR kit (Qiagen). The RT reaction was carried out at 50 °C for 30 min for the first cDNA strand synthesis. The PCR reaction was started at 95 °C for 15 min to initiate the hot start for HotStarTaq DNA polymerase provided in the RT-PCR mixture, followed by 40 cycles of 94 °C for 30 s, 55 °C for 30 s, 72 °C for 1 min, then 10 min elongation at 72 °C and a 4 °C hold for short term storage. The RT-PCR products were used directly (without purification) as templates in nested-PCR to amplify antibody variable regions with pfx50 DNA polymerase (Invitrogen). The PCR conditions for amplifying heavy, kappa, light chain variable regions remained the same: 94 °C for 2 min, then 10 cycles of 94 °C for 30 s, 50 °C for 30 s, 68 °C for 1 min, followed by 30 cycles of 94 °C for 30 s, 60 °C for 30 s, 68 °C for 1 min, then 7 min elongation at 68 °C and 4 °C hold for short term storage. The nested PCR products were then used as templates in overlapping PCR to connect antibody light and heavy chain. Overlapping PCR condition for connecting light chain (Kappa or Lambda) and heavy chain with linker is: 94 °C for 2 min, then 10 cycles of 94 °C for 30 s, 60 °C for 30 s, 68 °C for 2 min, followed by 30 cycles of 94 °C for 30 s, 65 °C for 30 s, 68 °C for 2 min, then 7 min elongation at 68 °C and 4 °C hold for short term storage. The overlapping PCR product was sub-cloned with infusion HD cloning kit (Clontech) following the manufacturer's instructions and sequenced. The human B cell cloning primers are provided in Supplementary Table 1.

**Alignment of the RB1 variable region with human germline.** The amino acid sequence of the RB1 antibody was aligned to the human antibody germline sequence using the IgBlast (https://ftp.ncbi.nih.gov/blast/executables/igblast/release/LATEST) program and the antibody complementarity-determining region (CDR) of the heavy chain (shown as RB1_VH) and the light chain (shown as RB1_VK) were determined according to Kabat numbering system.

**Recombinant antibody production.** The naturally paired heavy and light chain variable region sequences were synthesized and were then subcloned into the pTT5 vector for CHO-3E7 (National Research Council Canada cat.#11992) cell expression. CHO-3E7 cells were grown in suspension in serum-free FreeStyle CHO Expression Medium (Life Technologies). The recombinant plasmids encoding heavy and light chains of each antibody were transiently co-transfected into CHO-3E7 cells and cultured for 6 days. Cell culture supernatant was collected, centrifuged and filtered. Filtered supernatant was loaded onto a Protein A CIP column (GenScript). After washing and elution, the eluate was buffer exchanged to 1× phosphate buffered saline (PBS) at pH 7.2.

**Assessment of binding affinities by SPR.** Binding of RB1 Fab to pre-fusion and post-fusion RSV F proteins was characterized by surface plasmon resonance using a Biacore T200 system (GE Healthcare, Piscataway, NJ). For prefusion protein, approximately 5000 RU of D25 antibody was loaded onto channels 1 and 2 of a Series S Protein A Sensor Chip (GE Healthcare) and DS-Cav1 was flowed over channel 2 under conditions that allowed capture of 50 RU of recombinant protein. Two-fold serial dilutions of monovalent RB1 Fab were flowed over both channels at 50 μl/min for 30 s, followed by a dissociation period of 300 s for 0.31 nm to 10 nM RB1 Fab concentrations or 1800 s for 20 nM RB1 Fab concentration. To analyze binding to post-fusion F, 5000 RU of Synagis antibody was loaded onto the protein A chip to capture 50 RU of recombinant post-fusion F protein. RB1 Fab was flowed in two-fold serial dilutions from 200 nM to 3.12 nM concentration followed by a 40 sec dissociation period. Protein A surfaces were regenerated for each cycle after two 20 s injections of 10 mM glycine buffer, pH 1.5 at a 30 μL/min flow rate. Sensorgrams were corrected for sensor background (channel 2–1) and double

referenced after subtraction of analyte using a blank (0 nM) RB1 Fab injection. Resulting data sets were fitted to a 1:1 Langmuir binding model using the Biacore T200 evaluation software (version 2.0) to determine the rate constant of association ($k_a$) and dissociation ($k_d$), and the equilibrium dissociation constant $K_D$.

**RSV microneutralization assay.** RB1 was diluted in EMEM containing 2% FBS (heat inactivated) starting at 10 μg/mL followed by 3-fold serial dilutions. RB1 dilutions were mixed with 100 pfu of RSV A (Long, ATCC cat. no: VR-26) strain or RSV B (Washington, ATCC cat. no: VR-1580) strain and incubated for 1 hr at 37 °C. Following the 1 h incubation, HEp-2 (ATCC cat. #CCL-23) cells were added to the plates at $1.5 \times 10^4$ cells/well. The plates were incubated at 37 °C for 3 days. Afterward, the cells were washed and fixed with ice-cold 80% acetone in PBS for 15 min. Mouse anti-RSV-F and anti-RSV-N mAbs (in-house generated, clone 143-F3-1B8 and 34C9, respectively) at 1.25 μg/mL each were added to the plates and incubated for 1 h at RT. Plates were then washed and biotinylated horse anti-mouse IgG (Vector Laboratories Cat. #BA-2000) at 1:200 dilution was added. One hour later, the plates were washed and developed by a dual channel near infrared detection (NID) system (Licor Odyssey Sa). Infrared dye-Streptavidin to detect RSV specific signal and two cell stains for assay normalization were added to the 96-well plates. Plates were incubated for 1 h in the dark, washed and dried in the dark for 20 min. Plates were then read on the Licor Aerius® Automated Imaging System utilizing a 700-channel laser for cell normalization and an 800-channel laser for detection of RSV specific signal. 800/700 ratios were calculated, and serum neutralizing titers were determined by four parameter curve-fit in GraphPad Prism.

**HMPV plaque reduction microneutralization assay.** Monoclonal test antibodies were diluted in OTI-MEM assay medium (Gibco 31985-070) in serial 3-fold dilutions. Antibody dilutions were mixed with 100 pfu of HMPV virus HMPV A (HMPV 16 Type A1 Strain: IA10-2003, ZepoMetrix, Ref# 0810161CF) or HMPV B (hMPV-8 Strain: Peru6-2003 B2, ZepoMetrix Ref# 0810159CF) and incubated for 1 h at 37 °C. Following the 1 h incubation, 50 μl of LLC-MK2 cells (ATCC # CCL-7.1) were added to the 96 well plates at a concentration of $1.2 \times 10^6$ cells/mL. The plates were incubated at 37 °C for 1 h, then centrifuged at $250 \times g$ for 10 min, covered with 1% methylcellulose overlay and incubated at 37 °C for 3 days. On day three, the overlay was removed from the wells using a 12-channel vacuum plate washer and the remaining cells/viral mixture was fixed with 10% formalin. The plates were incubated at ambient temperature for 30 min and fixative was removed. The plates were washed six times with PBS-0.05% Tween-20, blocked (Odessey catalog #927-4000), for 30 min. The blocking buffer was removed and the plates were incubated with anti-HMPV mab (EMD Millipore Cat. #MAB80124) 1:1000 in blocking buffer for 1 h. Plates were washed six times with PBS-0.05% Tween-20, and the antibody was detected using IgG Alexa 488 conjugated secondary antibody (Invitrogen #A11017) 1:500. The plates were incubated for 1 h at ambient temperature and read using an EnSight plate reader (Perkin Elmer).

**Analyses of RSV sequences.** Complete RSV F glycoprotein sequences from 3600 submissions were obtained from GenBank for analysis (https://www.ncbi.nlm.nih.gov/genbank/; accessed April 2019). Out of these 3,058 contained the complete extracellular domain and had no ambiguous amino acids in antigenic site IV. Genbank entry AHA61614 was excluded from analysis due to an atypically large degree of sequence divergence from other RSV A and RSV B sequences according to our own analysis and as reported in Souza, et al.[42] The sequences were aligned using a modified Seq2Logo program[43] and the amino acid frequency at each position was determined. RSV F glycoprotein sequences from an additional panel of 47 clinical isolates were sequenced.

**Dendrogram analysis of RSV F glycoprotein.** The extracellular domain of RSV F glycoprotein includes amino acids 27 through 529, per Uniprot annotation (The UniProt Consortium). Using the bioinformatics algorithm clustalW http://www.clustal.org/[44], a total of 391 extracellular domain sequences were clustered. The 391 sequences include the 345 unique GenBank sequences and the sequences from the panel of 47 clinical isolates. The clustered data was visualized using FigTree version 1.4.3 (http://tree.bio.ed.ac/software/figtree/).

**RSV clinical isolate collection and preparation.** A panel of 47 RSV clinical isolates containing amino acid changes in the F fusion glycoprotein were selected for the in vitro neutralization assay. The clinical RSV isolates were propagated in HEp-2 (ATCC cat. #CCL-23) cells; low-passage viral stocks were prepared for analysis. Titers of the low-passage clinical isolate viral stocks were determined using a plaque immunostaining assay. These viral stocks were subsequently used in the in vitro infection neutralization assay.

**Prophylactic in vivo cotton rat challenge model.** Five, 4–7-week old, female cotton rats (Sigmodon hispidus) per dose group (SAGE Animals, Boyertown, PA or Envigo, Somerset, NJ), were passively immunized, intramuscularly with RB1 (Fig. 4a–d), or with RB1, RB1-LALA (Fig. 4e), palivizumab, or palivizumab-LALA (Fig. 4f) For each experiment, antibody dilutions started at 2.5 mg/kg and were serially diluted 3-fold to 0.03 mg/kilogram (specifically, a total of five antibody dose

groups (5 animals/group) were tested: 2.5, 0.83, 0.27, 0.09, and 0.03 mg/kg). A single weight from averaging six random animals was used for dose calculation. Animals were administered antibody in the left quadricep, in a volume of 100 μL on day 1 under isoflurane anesthesia. A control group consisting of naïve animals was also included. Serum samples were collected for determining serum antibody concentration followed by intranasally inoculating animals with $10^5$ pfu of either RSV A2 (ATCC VR-1540) or RSV B 18537 (Washington, ATCC VR-1580) virus in 0.1 ml volume on day 2, approximately $24 \pm 2$ h post RB1 transfer, under anesthesia. Four days post inoculation, animals were sacrificed by CO2 inhalation and left lung lobes, as well as nasal turbinates, were removed and homogenized in 10 volumes of Hanks Balanced Salt Solution (Lonza) containing Sucrose Phosphate Buffer (SPG) on wet ice. Samples were clarified by centrifugation at $300 \times g$ for 10 min, aliquoted, flash frozen, and immediately stored frozen at $-70$ °C until thawed for viral titration. The animal studies were approved by Merck Sharp & Dohme Corp., a subsidiary of Merck & Co., Inc., Kenilworth, NJ, USA, Institutional Animal Care and Use Committee and conducted in accordance with animal care guidelines.

**RSV plaque assay for cotton rat challenge studies.** Cotton rat lung and nose homogenates were tested by plaque assay to determine plaque-forming units per gram (pfu/g) of tissue. Homogenates were diluted in serum free Williams E media and added, in duplicate to 24-well plates containing confluent HEp-2 (ATCC cat. #CCL-23) cells. Following a 1-h infection at 37 °C, the test samples were removed, and cells were overlaid with 1 mL of 0.75% methylcellulose in Williams E supplemented with 1.6% fetal bovine serum. After incubation at 37 °C for 5 days, the cells were fixed/stained with a crystal violet solution containing 5% glutaraldehyde. Plaques were counted visually using a dissecting microscope and used to calculate pfu/g.

**RSV mAb pK ELISA.** RSV pre-F or post-F protein was coated on 96 well plates at 2 μg/mL in $1 \times$ (PBS) overnight. Plates were washed and blocked with 3% milk in PBS containing 0.05% Tween 20 for 1 h. Serum samples were diluted 1:50 in blocking buffer, followed by 4-fold serial dilutions. One hundred (100) μL of each diluted serum sample and standard were added to the 96-well plates and incubated for two hours. For each assay, the standard control was the matched antibody. For example, for cotton rat studies dosed with RB1, serially diluted RB1 with a quantified concentration was used as the standard. Following incubation, plates were washed and 50 μL of horseradish peroxidase (HRP)-conjugated anti-human IgG (Invitrogen #62-7120) diluted 1:2000 in blocking buffer was added and incubated for 1 h. Plates were washed and developed with SuperBlu Turbo TMB and Stop Solution. The plates were read on a Molecular Devices VersaMax at OD 450 nm. Softmax software was used to perform 4-parameter curve fit to obtain the concentration of antibody in each sample.

**Calculation of cotton rat EC50.** The EC$_{50}$ values for each RSV strain were calculated as follows: Serum antibody concentration was measured for each animal on days 1 post administration using the immunoassay for antibody drug serum concentration described. RSV infectious titers were also determined on day 5 using the RSV plaque assay. The EC$_{50}$ values and 95% confidence intervals for the mAb in tissue were estimated by an inhibitory sigmoidal dose-response analysis with variable slope (GraphPad Prism® version 6).

Data were plotted as follows: The $x$-axis is the log10 transformed antibody serum concentration at day 1 in μg/mL where: 100 μg/mL = 2 log (μg/mL). The $y$-axis is the log10 transformed viral titer where :$1 \times 10^5$ pfu/g tissue $= 5$ log (pfu/g tissue) on day 5. The 4-parameter curve fits were fitted with the following constraints: the bottom of the curve was constrained to equal the limit of detection of the plaque assay, 2 log pfu/g tissue for lung and 1.6 log pfu/g tissue for nose, and the top of the curve constrained by the value of the titer average of the untreated controls.

**Statistical analysis of the cotton rat LALA antibody studies.** Statistical analysis was performed using two-tailed unpaired $t$-test with GraphPad statistical software (GraphPad Prism® version 6). To perform the analysis, data from each animal was used as a data point, as tested once each (5 animals per group in five dose group).

**Generation and sequencing of RSV escape mutants (MARMS).** The generation of RSV escape mutants in vitro was done similarly as reported in X. Zhao, et al and L. Tome, et al.[45,46]. Specifically, $3.5$–$4.5 \times 10^5$ HEp-2 (ATCC cat. #CCL-23) cells were plated per well in 6-well plates. Culture medium was removed after 24 h and cells were infected with RSV A2 strain at MOI of 10 (~$7.5 \times 10^6$ pfu/well) or B Washington strain ($1.7 \times 10^6$ pfu/well) at MOI of 4 in the presence of suboptimal concentrations of antibodies in 3 ml EMEM/2% FBS media. Cells and supernatants were harvested using a cell scraper when there was significant CPE. The harvests were transferred into 15 ml tubes and flash frozen in dry ice/ethanol bath, and then thawed in 37 °C water bath. The tubes were then centrifuged at $300 \times g$ at room temperature for 10 min. 1 ml of supernatant was used to infect new HEp-2 cells in the next round with 2 ml of fresh media containing the same or increased concentrations of mAbs. The virus was then harvested from wells containing the higher concentrations of mAbs and showing CPE and used in next round of

infection. The procedure was repeated until the mAb concentration reached 10 μg/ml with a total of 9 rounds of selection.

Escape mutants were single plaque picked under the microscope and used to infect HEp-2 cells in 24-well plates. The single-plaque infected HEp-2 cultures were applied in subsequent RNA extraction with Qiagen RNeasy Mini Kit (Qiagen, cat no: 74104) according to the manufacturer's instructions. The RNAs were then applied in the RT-PCR amplification of RSV F. A QIAGEN One-Step RT-PCR kit (Qiagen, cat no: 210212) was used in the RT-PCR reaction, the PCR condition is as following: 50 °C 30 min for RT reaction, then 95 °C 15 min incubation, followed by 40 cycles of 94 °C 30 s, 57 °C 30 s, 72 °C 2 min, then 10 min at 72 °C for elongation, and 4 °C hold for temporary storage. The forward primer sequence is 5'ATGGAGTTGCTAATCCTCAAAG, and the reverse primer sequence is 5' GTTACTAAATGCAAT ATTATTTATACCACTCAG. PCR products were sequenced and analyzed with Sequencher (Gene Codes Corp.) and Vector NTI (Invitrogen).

**ELISA for RB1 binding to MARMS-sequence mutant proteins.** MARM plasmid constructs containing desired point mutations were generated in the DS-Cav1 background by Genewiz (South Plainfield, NJ 07080). Plasmid DNA was used for transient transfection of Expi293F cells (ThermoFisher cat. #A14527). Expifectamine (ThermoFisher, cat #A14525) was used as the transfection reagent for transfections. Cells were transfected at a final cell density of $2.6 \times 10^6$ viable cells/ml. For each 30 ml of transfection cell culture, 30 μg of plasmid DNA and 80 μl of Expifectamine was used. Transfection was completed according to the manufacturer's recommendation for Expifectamine reagent. All cell cultures were incubated in 37 °C incubator with a humidified atmosphere, 8% $CO_2$ on an orbital shaker rotating at 125 rotations per minute with a throw of 1 inch. On day 3 and day 7 post transfection, 10 ml from each transfection was harvested by centrifugation at $2800 \times g$ at room temperature. Clarified supernatants were subjected to denaturing, reducing SDS-PAGE and western blot analysis using was used to confirm desired protein expression. Nunc C96 Maxisorp Nunc-Immuno Plates were then coated with RSV Pre-F protein or variants of the F protein, which incorporated the sequences changes found in the MARMS. Plates were incubated at 4 °C overnight, washed in PBS/0.05% Tween 20 and then blocked in PBS/0.05% Tween 20/3% non-fat milk at room temperature for 1 h. 50 μl per well of 3-fold serially titrated RB1 were then added and incubated at room temperature for 90 min. Plates were washed and 50 μl per well of HRP-conjugated goat anti-human IgG (1:2,000) (Southern Biotech Cat. # 2040-05) was added. After 60 min incubation at room temperature, plates were washed and 100 μl per well of SuperBlu-Turbo TMB Solution (ViroLabs) was added. Plates were incubated at room temperature for 5 min and the reaction was stopped by adding 100 μl per well of Stop Solution for TMB ELISA (ViroLabs). Plates were then read on VICTOR Multilabel Counter (Wallac/Perkin Elmer) at 450 nm.

**Epitope mapping with shotgun mutagenesis.** Shotgun mutagenesis epitope mapping of the RB1 binding site on the RSV F glycoprotein was performed by Integral Molecular, Inc. (Philadelphia, USA). The method utilized RSV F glycoprotein sequence derived from RSV-A2, National Center for Biotechnology Information (NCBI) reference number FJ614814. Alanine scanning mutagenesis of the expression construct for the RSV F glycoprotein targeted 368 surface-exposed residues identified from the crystal structures of both prefusion and post-fusion conformations of RSV F[17,18]. Each residue of interest was individually mutated to an alanine, with native alanine residues mutated to serine. The resulting 368 mutant RSV F glycoprotein expression constructs were sequence confirmed and arrayed into a 384-well plate with one mutation per well. Mutations within clones were identified as critical to the antibody binding epitope if they did not support reactivity to the test antibody (e.g., RB1), but supported reactivity of the control antibody (e.g., D25 or palivizumab). This counter-screen strategy involving a second anti-RSV antibody facilitates the exclusion of RSV F glycoprotein mutants that were misfolded or had an expression defect. The algorithms and interpretation of the shotgun mutagenesis were performed at Intergrol Molecular as described in Davidson et al.[28]

**Crystallization of RB1 and RSV Fusion glycoprotein.** To generate RB1/DS-Cav1 complex, RB1 mAb was cleaved into monovalent Fab using the Fab Preparation Kit (Pierce) per the manufacturer's instructions. Cleaved RB1 Fab was incubated with purified DS-Cav1 at 4 °C overnight at a 1 to 1.3 antigen to Fab ratio (weight/weight). Antigen/Fab complex was purified using size exclusion chromatography on a Superdex 200 column (GE Healthcare) in 50 mM HEPES, pH 7.5 and 300 mM NaCl. The RB1-DSCav1 complex was concentrated to 10 mg/ml in a buffer containing 20 mM HEPES pH 7.5, and 100 mM NaCl. Crystals were obtained in a condition containing 100 mM Tris pH 8.5, 12% PEG 8000, and 200 mM ammonium sulfate. Although crystals appeared readily, the initial crystals did not diffract to sufficient resolution. A combination of seeding, dehydration, and screening crystals yielded one that diffracted past 3.5 angstroms. The crystal was dehydrated for four days in a buffer containing 100 mM Tris pH 8.5 and 35% PEG 1500. The crystal was vitrified for data collection directly from the dehydration buffer by fast submersion into liquid nitrogen.

**Data collection and structure solution**. Data were collected at the Canadian Light Source, Beamline 8ID-1, using a single crystal. Collection was performed at 100 K, using an incident x-ray at 0.97490 Å. After the data were processed by using Autoproc[47], the phases were determined by molecular replacement using the program, Phaser[48], as implemented in the Phenix package[49]. The solution was obtained using the DS-CAV1 structure (PDB: 4MMT[16]) and a Fab (PDB code: 1HZH[50]), with the complementarity determining regions removed, as search models. Two copies of DS-Cav1 were found as well as one copy of Fab. Searches with two or more fabs did not yield solutions. After refinement using Buster[51], the other five copies of Fabs were generated by symmetry operations based on expected binding of one Fab to each monomer within the two trimers. The structure was refined by iterative cycles of rebuilding in Coot[52] and refinement with Buster or Phenix. The final structure has 94.9 % of residues in the favored region of the Ramachandran plot, 4.9% in the allowed region, and 0.2% in the outlier region. The structure of the complex between RB1 and RSV-F (DS-cav1) has been deposited in the wwPDB (code: 6OUS). Pymol (PyMOL Molecular Graphics System, Version 2.2 Schrödinger, LLC.) was used to generate molecular figures and perform structural alignments for modeling RB1 interactions with post-fusion F.

**Availability of materials**. The RB1 antibody is not publicly available. Samples and additional description will not be made available upon request at this time. Readers may contact the corresponding author request reagents or materials via a Material Transfer Agreement (MTA) which will be reviewed on a case-by-case basis.

**Reporting summary**. Further information on research design is available in the Nature Research Reporting Summary linked to this article.

## Data availability

Merck Sharp & Dohme Corp., a subsidiary of Merck & Co., Inc., Kenilworth, NJ, USA's data sharing policy, including restrictions, is available at http://engagezone.msd.com/ds_documentation.php through the EngageZone site or via email to dataaccess@merck.com. The source data underlying Fig. 1a–d, are provided as a Source Data file. The fusion protein sequences from the clinical isolates as well as accession numbers (when applicable), are provided as a Source Data file for Fig. 2a. The structure of the complex between RB1 and RSV-F (DS-cav1) has been deposited in the wwPDB (code: 6OUS).

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

## Acknowledgements

We thank Dr. Pedro Piedra, MD for providing us with RSV clinical isolates. We thank Dr. Guy Boivin, MD and Dr. Marie-Eve Hamelin, PhD (Laval University) for testing our antibody for the neutralization of a subset of the RSV clinical isolates. We would also like to thank Xi He for his work in the development of the RSV-GFP virus used in the screening neutralization assay, Michael Minnier for running the HMPV neutralization assay, and Dr. Antonios Aliprantis, MD, Ph.D. and Dr. Chung-Jr Huang, Ph.D (all employees of Merck Sharp & Dohme Corp, a subsidiary of Merck & Co., Inc., Kenilworth, NJ, USA) for critically reviewing the manuscript. The authors would like to acknowledge Drs. Barney Graham, MD. Ph.D., and Peter Kwong, Ph.D. (Vaccine Research Center, National Institute of Allergy and Infectious Diseases) for providing the plasmid encoding DS-Cav1. We thank Karyn Davis of Merck Sharp & Dohme Corp, a subsidiary of Merck & Co., Inc., Kenilworth, NJ, USA for editorial assistance. Use of the IMCA-CAT beamline 17-ID (or 17-BM) at the Advanced Photon Source was supported by the companies of the Industrial Macromolecular Crystallography Association through a contract with Hauptman-Woodward Medical Research Institute. This research used resources at the Industrial Macromolecular Crystallography Association Collaborative Access Team (IMCA-CAT) beamline 17-ID, supported by the companies of the Industrial Macromolecular Crystallography Association through a contract with Hauptman-Woodward Medical Research Institute. Research described in this paper was performed using beamline 08ID-1 at the Canadian Light Source, which is supported by the Canada Foundation for Innovation, Natural Sciences and Engineering Research Council of Canada, the University of Saskatchewan, the Government of Saskatchewan, Western Economic Diversification Canada, the National Research Council Canada, and the Canadian Institutes of Health Research. This study was funded by Merck Sharp & Dohme Corp., a subsidiary of Merck & Co., Inc., Kenilworth, NJ, USA.

## Author contributions

All authors contributed extensively to the work presented here. K.V., D.C., A.T., Z.C., K. C., H.S., A.F., L.Z., M.C., D.W. S.S. and D.D. designed various parts of the study or provided conceptual guidance. A.T., Z.C., K.C., C.C., S.P., P.C, R.S., S.T., M.C., D.G., B.L., M.E., G.H. J.R., J.G. and Z.W., designed assays and performed experiments. A.F. and H. S., collected and analyzed data. K.C. was the primary author with input on sections. All authors reviewed each draft.

## Additional information

**Competing interests:** A.T., Z.C., K.C, HP.S, C.C., A.F., L.Z., S. P., P.C.,R.S., S.T., M.C, B.L., M.E., J.R., S.S., J.G., D.W., Z.W., G.H., D.D. and K.V. are employees of Merck Sharp & Dohme Corp., a subsidiary of Merck & Co., Inc., Kenilworth, NJ, USA, and may hold stock in Merck & Co., Inc., Kenilworth, NJ, USA. D.C. and D.G. are former employees of Merck Sharp & Dohme Corp., a subsidiary of Merck & Co., Inc., Kenilworth, NJ, USA, and may hold stock in Merck & Co., Inc., Kenilworth, NJ, USA. D.C. is an employee of Sanofi Pasteur. D.G. is an employee of Janssen Research and Development.

