## [Peer Review File · Nature Communications]

Reviewers' Comments:

Reviewer #1:

Remarks to the Author:

Vora and colleagues describe the isolation and characterization of a monoclonal antibody mAb from human PBMC, a mAb specific for site IV of the RSV fusion protein. The manuscript shows that the newly isolated mAb has a strikingly high affinity for the pre-fusion F protein and high neutralizing titer. The authors defined the mAb binding site by analysis of the structure of the mAb-pre-F protein complex as well as isolation and sequence analysis of five monoclonal antibody resistant virus mutants. Further, in a cotton rat model, data are shown that the mAb has superior activity as a prophylactic treatment compared to palivizumab, which has been licensed for such treatment of at-risk infants.

The manuscript is a comprehensive biological and structural analysis of the newly isolated mAb RB1. The manuscript is well written with a large amount of data concisely presented. However, a number of site IV mAbs have been isolated and similarly characterized by many other groups including an antibody with similar affinity to F protein and neutralizing antibody titers (Mousa, et al Plos Pathogens 14 e1006837). Thus, this study is not novel but is a further characterization of antibody interactions with F protein site IV. What is an important contribution of this study is analysis of the prophylactic activity of the mAb with a comparison with palivizumab, information that has not been previously presented in other studies, to this reviewer's knowledge.

Also, a contribution of this manuscript is a more detailed and comprehensive analysis of the variation, or lack thereof, in the sequence of site IV across a large number of clinical isolates confirming previous conclusions that this site sequence is highly conserved, an important point in development of prophylactic antibodies. This manuscript demonstrates that mAb resistant virus mutants have reduced growth in tissue culture, a result consistent with the conclusion that the site IV sequence does not tolerate variation. This point is negatively impacted, however, by the quality of Figure 2 which is virtually unreadable as presented.

Missing from this manuscript is a sequence of the L and H antigen binding domains (H CDR and L CDR) of the RB1 antibody, a piece of data routinely presented in other similar studies (for example, Gilman, et al Sci. Immunology 1: eaaj1879; Mousa, et al Plos Pathogens 14 e1006837; Mousa, et al Nature Micro. 2: 16271). Also missing is the V gene usage which has been routinely presented in other similar studies.

Missing also is an analysis of competition for binding to pre-F protein by other site IV antibodies, data which is usually presented in such studies. This information may be informative, since it has been shown by Mousa, et al, (Mousa, et al Plos Pathogens 14 e1006837) that different site IV antibodies have different "binding modes" at least to the post fusion F. In this context, at least modeling of the binding of the antibody to post F would be informative.

Other issues:

1. As noted above, Figure 2 needs revision.
2. Figure S1: It would be informative to present the binding of other mAb, such as 3M3, to the MARM.
3. Figure S2: The color code is not presented thus interpretation of this figure is impossible. In addition, the Y axis is unclear. The meaning of the number 2000 should be described in the legend.
4. There is no apparent statistical analysis of data in figure 4

Reviewer #2:

Remarks to the Author:

Tang et al. report the isolation and detailed characterization of a recombinant human monoclonal IgG (RB1), derived from memory B cells, that can neutralize RSV A and B viruses. The antibody recognizes

site IV and has some preference for binding to F in the prefusion conformation. Co-crystal structure analysis was performed, which revealed that the CDRs of mAb RB1 Fab interact with amino acid residues in F that are conserved in RSV A and B viruses. In vitro RB1 escape viruses could be isolated. The predicted F sequence from these escape viruses carried amino acid residue changes at positions 443 and/or 446 and the escape viruses were slightly attenuated in vitro. Prophylactic intramuscular administration of RB1 appears to be able to reduce RSV A and B virus loads in the nose and lungs of experimentally challenged cotton rats.

Site IV specific mouse and human mAbs have been described before. In this respect, the presented work is not entirely novel. However, the large panel of clinical RSV isolates tested, the co-crystallography data, the dose-escalation data from the cotton rat challenge experiment, and the ongoing clinical development of an RB1 derivative in pre-term and full-term infants, warrant consideration of the work for possible publication in Nature Communications.

Major remarks

1. The cotton rat experiment is poorly described. The presentation of the outcome on RSV A and B plaque reduction in figure 4 is also confusing.

In the Methods section, the number of groups of 5 animals should be clearly described. Probably 3 doses were tested of RB1, RB1LALA and palivizumab for challenge with RSV A2 or RSV B 18537? The outcome of the plaque assay should be represented as the log of the number of plaques per gram of tissue (lung or nasal turbinates) of each animal in each group. In addition, it is important to perform RT-qPCR on viral genomic RNA in all the samples to rule out that RB1 Ab in the tissue samples contributes to plaque reduction ex vivo.

2. The apparent preference of RB1 for pre-F over post-F appears to be much higher based on the SPR experiment compared with the Elisa outcome (Fig. 1 and Table 1). The SPR experiment compares capturing with D25 of pre-F with capturing of post-F with synagis. D25 is remote from site IV whereas the epitope of synagis (site II) is next to it. Full access of RB1 to post-F (and pre-F) can therefore be hampered by synagis capture. The SPR should be repeated with pre- and post-F directly coated to the chip. Alternatively, capture of pre-F could also be done with synagis instead of D25.

3. It is custom to provide the crystallography data collection and refinement statistics. This information should be provide. The authors should also comment/speculate on the preference (however see comment 2 above) for pre-F based on the co-crystal structure analysis. In addition, it would be helpful for the reader if the position of the contact residues between RB1 and pre-F are also shown in the post-F conformation. Line 179 and elsewhere in the manuscript: Glu161 and Ser182 are part of F2, not F1. Please correct.

4. RSV mAb pk Elisa (line 447). Please specify the standard(s) and the dilution and source of the anti-human IgG-HRP conjugate that was used.

5. In the discussion, the authors compare the neutralizing potency with previously described F-specific monoclonal antibodies and conclude that the neutralization potency of RB1 is "among the highest potencies". However, the IC50 values of RB1 were determined based on an in house developed RSV microneutralization assay. The assay that is used is unusual in the field and the authors use in house developed reference mAbs instead of more established comparators such as palivizumab or D25. Typically a RSV plaque reduction assay is used to determine the neutralization potency of a mAb. Therefore, it is unjustified to compare neutralization potencies. Either these statements should be removed or the outcome of the microneutralization assay with palivizumab or D25 should be added in Fig 1A.

Other remarks

1. Line 40: RSV is a member of the Pneumoviridae.

2. Please improve table 2: the amino acid sequence that is shown at first glance can be considered a continuous linear peptide sequence in RSV F.

3. Line 272: site IV spans residues 422 to 468. Please correct.

4. The IgG isotype of RB1 should be mentioned.

Reviewer #3:

Remarks to the Author:

The manuscript from Tang et al reports the isolation and characterization of a human monoclonal antibody (RB1) that binds to the RSV F glycoprotein. RB1 was isolated from human PBMCs via B cell sorting with postfusion RSV F as the antigen bait. SPR studies demonstrated that RB1 binds to both prefusion and postfusion RSV F, but the affinity to prefusion F was much higher. RB1 potently neutralized a diverse set of clinical RSV isolates from both A and B subgroups. Shotgun mutagenesis, X-ray crystallography, and antibody-escape mutations all mapped the RB1 epitope to antigenic site IV. Analysis of 3000+ RSV F sequences revealed that antigenic site IV is very well conserved, with only a few sequences containing a substitution in or near the RB1 epitope. In the cotton rat challenge model, RB1 was shown to have prophylactic efficacy that is independent of antibody effector functions. A variant of RB1 with Fc mutations to extend half-life is in development for passive prophylaxis.

The manuscript is clearly written and the data are presented well. The majority of the conclusions are well supported by the data. RSV monoclonal antibody development is at a critical period, with a recent Phase III failure from Regeneron and a highly anticipated Phase II result from MedImmune. The RB1 antibody targets an epitope distinct from those two, and its isolation and characterization will be of interest to many in the field. However, the manuscript could be improved, and specific comments are provided below.

Major comments

1. There are no statistics presented for any of the data. There are also no error bars for the neutralization panel shown in Figure 2B or the efficacy data in Figure 4.
2. Given the variability of neutralization assays within the RSV community, it would be helpful if a well-studied antibody, such as palivizumab or D25 (which the authors used in Fig S1B), were included in the neutralization panel as comparators.
3. The SPR curves need to be included, as well as the fit to a 1:1 Langmuir binding model, so that their quality can be assessed.
4. The X-ray crystal structure is not particularly well built as judged by the Ramachandran outliers and low percentage of residues in favored regions. Generally >95% of residues should be within favored regions of the Ramachandran plot, but the PDB validation report shows that the majority of chains have between 83% and 92% of residues within favored regions. Given that there are numerous structures of RSV F deposited with better geometry, those structures could be used for reference-model restraints.

Other comments

1. Line 215 states that MARMS were identified using an RSV A Long strain, but Table 3 and subsequent text states that RSV A2 strain was used.
2. Line 296 states that RB1 contacts two adjacent protomers. Please provide the surface area buried on each protomer.
3. Line 303 states that RB1 does not neutralize hMPV, however no data are provided. Please provide the data to support this statement.
4. Lines 306-307 states REGN2222 failed to meet its primary endpoint in a phase III clinical trial, "further highlighting the importance of resistant variants in the RSV field". What is the connection between the Phase III trial and importance of resistant variants?
5. Software programs are generally capitalized.
6. Line 40: 'Syncytia' should be 'syncytial'. And RSV does not belong to the paramyxoviridae family, but rather the pneumoviridae family.

Reviewers' comments:

Reviewer #1 (Remarks to the Author):

Vora and colleagues describe the isolation and characterization of a monoclonal antibody mAb from human PBMC, a mAb specific for site IV of the RSV fusion protein. The manuscript shows that the newly isolated mAb has a strikingly high affinity for the pre-fusion F protein and high neutralizing titer. The authors defined the mAb binding site by analysis of the structure of the mAb-pre-F protein complex as well as isolation and sequence analysis of five monoclonal antibody resistant virus mutants. Further, in a cotton rat model, data are shown that the mAb has superior activity as a prophylactic treatment compared to palivizumab, which has been licensed for such treatment of at-risk infants.

The manuscript is a comprehensive biological and structural analysis of the newly isolated mAb RB1. The manuscript is well written with a large amount of data concisely presented. However, a number of site IV mAbs have been isolated and similarly characterized by many other groups including an antibody with similar affinity to F protein and neutralizing antibody titers (Mousa, et al Plos Pathogens 14 e1006837). Thus, this study is not novel but is a further characterization of antibody interactions with F protein site IV. What is an important contribution of this study is analysis of the prophylactic activity of the mAb with a comparison with palivizumab, information that has not been previously presented in other studies, to this reviewer's knowledge.

Also, a contribution of this manuscript is a more detailed and comprehensive analysis of the variation, or lack thereof, in the sequence of site IV across a large number of clinical isolates confirming previous conclusions that this site sequence is highly conserved, an important point in development of prophylactic antibodies. This manuscript demonstrates that mAb resistant virus mutants have reduced growth in tissue culture, a result consistent with the conclusion that the site IV sequence does not tolerate variation.

This point is negatively impacted, however, by the quality of Figure 2 which is virtually unreadable as presented.

We agree and have provided a higher resolution figure for publication (Figure 2A).

Missing from this manuscript is a sequence of the L and H antigen binding domains :

(H CDR and L CDR) of the RB1 antibody, a piece of data routinely presented in other similar studies (for example, Gilman, et al Sci. Immunology 1: eaaj1879; Mousa, et al Plos Pathogens 14 e1006837; Mousa, et al Nature Micro. 2: 16271). Also missing is the V gene usage which has been routinely presented in other similar studies.

The sequence has now been included as Supplemental Figure 1. The H and L chain sequence has been published earlier as a part of patent (pat #US9,963,500 B2 May 8th2018); furthermore, the sequence was also deposited as a part of the crystal structure validation report. The germline VH and VL gene belong to VH3-49*04 and KV1D-13*01 family respectively. H- and L-chain J regions belong to JH6*02 and JK5*01 respectively and the D segment was identified as D4-23*. This information has been added as a Supplemental Figure 1 to result section lines 116-120.

Missing also is an analysis of competition for binding to pre-F protein by other site IV antibodies, data which is usually presented in such studies. This information may be informative, since it has been shown by Mousa, et al, (Mousa, et al Plos Pathogens 14 e1006837) that different site IV antibodies have different “binding modes” at least to the post fusion F. In this context, at least modeling of the binding of the antibody to post F would be informative.

We agree that comparison of RB1 to the antibodies reported in Mousa, et al Plos Pathogens 14 e1006837 would be informative. Unfortunately, the sequences of the site IV antibodies from Mousa et al manuscript are not available in public domain to the best of our knowledge. However, we have modeled binding modes of RB1 compared to 101F for which the crystal structure was published earlier, in our manuscript (Figure 3C). We sincerely feel that the direct competition experiments to the antibodies reported in Mousa et al is out of scope for the current manuscript and could be the topic of follow-up manuscripts. We will reach out to Professor Jim Crowe (senior author on the Mousa et al publication) for the sequences of the site IV antibodies for the proposed competition experiments in the future. We have added the modeling of RB1 to post-fusion F as figure 3D/E in the revised manuscript.

Other issues:

1. As noted above, Figure 2 needs revision.

We have obtained a higher quality figure for publication (Fig 2A).

2. Figure S1: It would be informative to present the binding of other mAb, such as 3M3, to the MARM. We agree that this would be informative but would be out of scope for the current manuscript and may be the topic of an upcoming manuscript. As stated earlier we will reach out to Professor Jim Crowe to conduct the suggested experiments.

3. Figure S3: The color code is not presented thus interpretation of this figure is impossible. In addition, the Y axis is unclear. The meaning of the number 2000 should be described in the legend.

Thank you for pointing out this omission. We have revised the figure (now supplemental Figure 4) and the Y axis is now clearly described along with a description of the color coding in the legend.

4. There is no apparent statistical analysis of data in figure 4.

Statistical analysis result was added to lines 283-285 and the methods to lines 567-570.

Reviewer #2 (Remarks to the Author):

Tang et al. report the isolation and detailed characterization of a recombinant human monoclonal IgG (RB1), derived from memory B cells, that can neutralize RSV A and B viruses. The antibody recognizes site IV and has some preference for binding to F in the prefusion conformation. Co-crystal structure analysis was performed, which revealed that the CDRs of mAb RB1 Fab interact with amino acid residues

in F that are conserved in RSV A and B viruses. In vitro RB1 escape viruses could be isolated. The predicted F sequence from these escape viruses carried amino acid residue changes at positions 443 and/or 446 and the escape viruses were slightly attenuated in vitro. Prophylactic intramuscular administration of RB1 appears to be able to reduce RSV A and B virus loads in the nose and lungs of experimentally challenged cotton rats.

Site IV specific mouse and human mAbs have been described before. In this respect, the presented work is not entirely novel. However, the large panel of clinical RSV isolates tested, the co-crystallography data, the dose-escalation data from the cotton rat challenge experiment, and the ongoing clinical development of an RB1 derivative in pre-term and full-term infants, warrant consideration of the work for possible publication in Nature Communications.

Major remarks

1. The cotton rat experiment is poorly described. The presentation of the outcome on RSV A and B plaque reduction in figure 4 is also confusing.

In the Methods section, the number of groups of 5 animals should be clearly described. Probably 3 doses were tested of RB1, RB1LALA and palivizumab for challenge with RSV A2 or RSV B 18537?

Thank you for the comments. We have now revised the methods description by adding more details to the methods section, clearly stating that five doses per antibody and five animals per each dose for all cotton rat experiments (challenge virus A or B). The challenge strains were RSV A2 and RSV B18537.

The outcome of the plaque assay should be represented as the log of the number of plaques per gram of tissue (lung or nasal turbinates) of each animal in each group.

We agree with the reviewer and we have presented the data as scientific numerals (rather than \log_{10}) as suggested. The dotted line around each curve represents the 95% confidence interval.

In addition, it is important to perform RT-qPCR on viral genomic RNA in all the samples to rule out that RB1 Ab in the tissue samples contributes to plaque reduction ex vivo.

RT-qPCR tends to give false positives more often than plaque assays due to amplification of the target sequence from inactivated viruses and hence it is more common in the field to use plaque assays which readout live virus. Zhu et al (Science translational medicine) have reported similar cotton rat studies with palivizumab and MEDI8897 as plaque assays pfu/gram of tissue. Appended below is a correlation plot between the PCR methods and plaque assay from a BAL sample from non-human primate experiment Panel A (internal, unpublished data). As can be seen from the Figure panel B below, the PCR assays consistently gave us >2 log higher viral titers than the plaque assay. It is interesting to note that the patterns/trends of the curves remain similar. Furthermore, since EC_{50} reads antibody concentrations leading to ~2 log reduction in viral loads relative to the starting loads in control animals, we don't see these values changing upon use of different methods.

Additionally, we have tested the interference of antibody in the plaque assay (as shown below in the three-panel Figure). Briefly, cotton rat lung lysates dosed with different amounts of palivizumab were spiked with different amounts of RSV A2 virus and their titers measured in a plaque assay (B and C). Alternatively, cotton rat animals were challenged with RSV virus and then the plaque assay performed on lung lysates (Figure A). Figure B and C clearly demonstrate that palivizumab in the lung did not interfere in the spiked virus plaque assay. Palivizumab was active *in vivo* as it caused reduction of RSV plaques in a *in vivo* RSV challenge study Figure A. These figures are for reviewer's information only and not to be included in the manuscript.

[redacted]

We hope that the above experiments are enough to allay the reviewer's concerns on the choice of plaque assay to read out the efficacy of RB1 in a passive immunization/challenge cotton rat model.

2. The apparent preference of RB1 for pre-F over post-F appears to be much higher based on the SPR experiment compared with the Elisa outcome (Fig. 1 and Table 1). The SPR experiment compares capturing with D25 of pre-F with capturing of post-F with synagis. D25 is remote from site IV whereas the epitope of synagis (site II) is next to it. Full access of RB1 to post-F (and pre-F) can therefore be hampered by synagis capture. The SPR should be repeated with pre- and post-F directly coated to the chip. Alternatively, capture of pre-F could also be done with synagis instead of D25.

We agree with the reviewer that the SPR experiment indicates a stronger RB1 bias to preF binding than the ELISA. However, we would argue that this could be explained by intrinsic assay differences rather than by binding competition between RB1 and palivizumab antibodies. Most importantly, we measure K_D values of RB1 Fab by SPR while full length antibody is used in the ELISA. The avidity effect could explain the significant RB1-postF binding detected by ELISA. The other important point to note is that SPR is done under a flow condition whereas the ELISA assay is static in a well. This discordance could account for the differences seen between SPR and ELISA. We have added the clarifying explanation in the revised manuscript on lines 138-144.

As the reviewer suggests, we first thought to run the SPR experiment using preF and postF bound directly to the chip. However, we abandoned this idea since postF-like structures are present even in fresh preparations of DS-Cav1 which would affect the measurements significantly. These postF-like “contaminants” are effectively eliminated by binding to D25. (Figure 3 of Flynn et al, PLoS ONE (2016)11(10): e0164789. <https://doi.org/10.1371/journal.pone.0164789>)

While the palivizumab and RB1 epitopes are relatively close, they do not compete for the binding as shown below via Biacore experiment (experiment done with full length antibodies). The analysis was done with capture of pre- or post-F protein with Synagis followed by addition of RB1. These data indicate that both Synagis and RB1 can bind simultaneously Pre and Post-F and hence there is limited competition between these antibodies. Additionally, we have confirmed these data in a competition ELISA (data not shown). Furthermore, since the exact binding sites for both antibodies are known and the sites are non-overlapping it is very unlikely that the competition between Synagis and RB1 binding to trimeric F proteins (pre- and post-F) is likely the reason for the faster off rates observed for RB-1 on post-F protein.

The K_{on} rates of RB1 Fab binding to preF and postF conformations are in the same order of magnitude (7.3×10^6 vs $1.1 \times 10^6 \text{ M}^{-1} \text{ s}^{-1}$), suggesting no significant steric hindrance in the postF conformation, as

expected when binding a trimeric molecule. In contrast, the K_{off} are significantly different (1.6×10^{-4} vs 0.15 s^{-1}), indicating that the different dissociation rates are the main cause for the preF binding bias.

3. It is custom to provide the crystallography data collection and refinement statistics. This information should be provide.

Thank you for the suggestion. We have provided a Table of Crystallographic Statistics in our resubmission as Supplemental Table 1.

The authors should also comment/speculate on the preference (however see comment 2 above) for pre-F based on the co-crystal structure analysis. In addition, it would be helpful for the reader if the position of the contact residues between RB1 and pre-F are also shown in the post-F conformation.

To address this question, an additional panel has been added Figure 3 (D/E). The figure highlights the anticipated differences in binding between post fusion F and the contacts identified in the structure with prefusion F. Details of the differences have also been added to the text as follows (lines 196-209).

The lower binding affinity to post-fusion F relative to prefusion F, as determined by SPR, can be rationalized by anticipated changes to the structure upon transition between the forms. The post-fusion F structure (pdb code: 3rrr) was superposed on the prefusion F structure based on residues 425 to 447, which encompass 12 of the 14 interacting residues on F. In Figure 3D, the structure with prefusion F is shown on the left with in ribbon with RB1 in cartoon (cyan and purple). The interacting residues from the neighboring monomer of F, Glu 161 and Ser 182, are shown in sticks in grey. On the right side of the Figure is the superposed post-fusion F protein but depicting that the neighboring monomer is no longer near the antibody. Additionally, a strand (shown in cartoon depiction) spanning residues 464-470 is positioned within 4 angstroms of the antibody. In the post-fusion conformation, the strand swings up towards the fusion bundle. Figure 3E shows the overlay from a farther perspective. Residues that form the epitope are depicted as spheres. On the right, the shift in the neighboring monomer and residues 161 and 182 now form the fusion bundle. Of the 1006 \AA^2 interface made between the antibody and the antigen, 191 \AA^2 is made to the neighboring monomer, which is no longer available in the post-fusion form.

Line 179 and elsewhere in the manuscript: Glu161 and Ser182 are part of F2, not F1.

Thank you for your comment. We have changed the F2 and F1 nomenclature in the paper. What we meant by this was the neighboring protomers of the fusion protein within the trimer. However, we agree after seeing your comment that this could be confused with the F1/F2 nomenclature for the full-length protein. We have changed our wording to call them protomer 1 and protomer 2 for discussions around the binding of the antibody across adjacent protomers.

4. RSV mAb pk Elisa (line 447). Please specify the standard(s) and the dilution and source of the anti-human IgG-HRP conjugate that was used.

This detail was added in line 548-551.

5. In the discussion, the authors compare the neutralizing potency with previously described F-specific monoclonal antibodies and conclude that the neutralization potency of RB1 is “among the highest potencies”. However, the IC50 values of RB1 were determined based on an in house developed RSV microneutralization assay. The assay that is used is unusual in the field and the authors use in house developed reference mAbs instead of more established comparators such as palivizumab or D25. Typically a RSV plaque reduction assay is used to determine the neutralization potency of a mAb. Therefore, it is unjustified to compare neutralization potencies. Either these statements should be removed or the outcome of the microneutralization assay with palivizumab or D25 should be added in Fig 1A. we can add pali to figure.

The IC50 of RB1 is determined using the LiCOR assay. The LiCOR has been an established method used to determine neutralizing titers for several different virus.

1. Wang et al., Sci. Transl. Med. 8, 362ra145 (2016) DOI: 10.1126/scitranslmed.aaf9387;
2. Chen et al., PLoS One. 2016; 11(6): e0156798 DOI: 10.1371/journal.pone.0156798
3. Chai et al., PLoS Pathog 12 (6): e1005702. doi:10.1371/journal.ppat.1005702
4. Weldon, S.K. et al., J Virol Methods, 168(1-2), pp 57-62. DOI: 10.1016/j.jviromet.2010.04.016
6. Lin, Y.C.J. et al., J Virol, 82(12), pp 5922-32. DOI: 10.1128/JVI.02723-07
7. Sklan E.H. et al., J Virol, 82(20), pp 11096-105. DOI: 10.1128/JVI.01249-07
8. Counihan, N.A. et al., J Virol Methods, 133(1), pp 62-9. DOI: 10.1016/j.jviromet.2005.10.023).

Furthermore, we do not see much difference between the LiCoR assay and plaques assay IC₅₀ given a 3X dilution scheme (Table below). Additionally, using both assays i.e. LiCoR and Plaque assay we get similar potencies for RB1 [redacted]. We have included palivizumab data in Figure 1 A as requested. Please note that the IC₅₀ presented in the table below and in the graphs were 2 different experiments and slightly different but within the experimental variation. We choose as a principle not to present direct potency comparisons to other competitor antibodies in clinical development in the manuscript to avoid the perception that one antibody is inferior or superior to the other. We have used D25 in an epitope mapping experiment as a tool to ensure proper folding of the mutant proteins, but not in the context of potency comparison in the current manuscript.

mAb	In Vitro neutralization IC ₅₀ ng/ml RSV A Long	
	LiCor Assay	Plaque Assay
RB1	3.8	10.6
[redacted]	[redacted]	[redacted]
Palivizumab	212	329.3

[redacted]

Other remarks

1. Line 40: RSV is a member of the Pneumoviridae.

Thank you, this was edited.

2. Please improve table 2: the amino acid sequence that is shown at first glance can be considered a continuous linear peptide sequence in RSV F.

The Figure has been edited to indicate breaks in the sequence.

3. Line 272: site IV spans residues 422 to 468. Please correct.

This has been corrected.

4. The IgG isotype of RB1 should be mentioned

This is listed (IgG1) in line 99 in the original submission. In this resubmission, we added it to line 120 as well to be clearer.

Reviewer #3 (Remarks to the Author):

The manuscript from Tang et al reports the isolation and characterization of a human monoclonal antibody (RB1) that binds to the RSV F glycoprotein. RB1 was isolated from human PBMCs via B cell sorting with postfusion RSV F as the antigen bait. SPR studies demonstrated that RB1 binds to both prefusion and postfusion RSV F, but the affinity to prefusion F was much higher. RB1 potently neutralized a diverse set of clinical RSV isolates from both A and B subgroups. Shotgun mutagenesis, X-

ray crystallography, and antibody-escape mutations all mapped the RB1 epitope to antigenic site IV. Analysis of 3000+ RSV F sequences revealed that antigenic site IV is very well conserved, with only a few sequences containing a substitution in or near the RB1 epitope. In the cotton rat challenge model, RB1 was shown to have prophylactic efficacy that is independent of antibody effector functions. A variant of RB1 with Fc mutations to extend half-life is in development for passive prophylaxis.

The manuscript is clearly written and the data are presented well. The majority of the conclusions are well supported by the data. RSV monoclonal antibody development is at a critical period, with a recent Phase III failure from Regeneron and a highly anticipated Phase II result from MedImmune. The RB1 antibody targets an epitope distinct from those two, and its isolation and characterization will be of interest to many in the field. However, the manuscript could be improved, and specific comments are provided below.

Major comments

1. There are no statistics presented for any of the data. There are also no error bars for the neutralization panel shown in Figure 2B or the efficacy data in Figure 4.

We have added error bars to Figure 2B, and statistical analysis for Figure 4 in which the LALA is compared to the parental antibodies RB1 or palivizumab. For Figure 4 A-D, 95% confidence intervals are shown as dotted lines in each plot and each animal result is shown as a point on the graph.

2. Given the variability of neutralization assays within the RSV community, it would be helpful if a well-studied antibody, such as palivizumab or D25 (which the authors used in Fig S1B), were included in the neutralization panel as comparators.

We have added palivizumab to Figure 1, and provided the D25 data for reviewer's consideration as shown in response to reviewer 2 point 5 above.

3. The SPR curves need to be included, as well as the fit to a 1:1 Langmuir binding model, so that their quality can be assessed.

SPR curves have been included as Supplemental Figure 2. These curves have been fit 1:1 to the Langmuir model and this has been included in the Figure legend.

4. The X-ray crystal structure is not particularly well built as judged by the Ramachandran outliers and low percentage of residues in favored regions. Generally >95% of residues should be within favored regions of the Ramachandran plot, but the PDB validation report shows that the majority of chains have between 83% and 92% of residues within favored regions. Given that there are numerous structures of RSV F deposited with better geometry, those structures could be used for reference-model restraints.

We thank the reviewer for their suggestion to use reference-model restraints. Through multiple rounds of refinement using reference models and removing regions of the structure where electron density was

weakly defined, the model has been refined to where 95% of the residues are in favored regions. The validation report is included with the resubmission.

Other comments

1. Line 215 states that MARMS were identified using an RSV A Long strain, but Table 3 and subsequent text states that RSV A2 strain was used.

The correct virus is A2, this has been corrected. (line 232)

2. Line 296 states that RB1 contacts two adjacent protomers. Please provide the surface area buried on each protomer.

Based on analysis using PISA, the interface between the antibody and the main monomer of RSV-F is 815 Å². The interface with the adjacent monomer is 191 Å².

3. Line 303 states that RB1 does not neutralize hMPV, however no data are provided. Please provide the data to support this statement.

In a hMPV A and hMPV B plaque neutralization assay we demonstrate that RB1 cannot neutralize hMPV A or B strains. In contrast, the positive control mAb MPE8 clearly demonstrated potent neutralization of hMPV A and B strains with IC50's 21.8 and 42.1 ng/ml respectively. We have added the data to Figure 1C to support our comment in the manuscript. Please note that there are some replicate outliers in the RB1 neutralization assay for the HMPV B2 strain. We have some bounce in that assay, however, it was random in some replicates, as provided in a source data file with our resubmission, and RB1 does not have any activity against HMPV.

4. Lines 306-307 states REGN2222 failed to meet its primary endpoint in a phase III clinical trial, "further highlighting the importance of resistant variants in the RSV field". What is the connection between the Phase III trial and importance of resistant variants?

In the RSV 2018 meeting (Asheville, NC), a presentation was given regarding the phase III trial results of REGN2222. In this presentation, it was revealed that an RSV B strain was in circulation at the time of the

trial, which was resistant to the antibody. This variant was also found in the placebo group; hence it was not attributed to emergence from treatment, but rather a naturally circulating strain.

5. Software programs are generally capitalized.

Thank you for your comment, this has been edited accordingly.

6. Line 40: 'Syncytia' should be 'syncytial'. And RSV does not belong to the paramyxoviridae family, but rather the pneumoviridae family.

Thank you for your comment, these errors have been edited in our revised manuscript (line 40).

Reviewers' Comments:

Reviewer #1:

Remarks to the Author:

This manuscript is a revised version of a previously submitted manuscript. The authors have satisfactorily addressed all concerns of this reviewer. It also appears that issues raised by other reviewers have been addressed appropriately.

One small comment is that in supplemental figure 4, there appears to be a stray star over residue 120. If this is intentional, its meaning is not clear.

Reviewer #2:

Remarks to the Author:

The authors have addressed all raised comments adequately.

Reviewer #3:

Remarks to the Author:

The authors have adequately addressed my concerns and the manuscript is much improved, as is the quality of the structure. Please do add units (angstroms, degrees, etc.) to the crystallographic table though.

REVIEWERS' COMMENTS:

Reviewer #1 (Remarks to the Author):

This manuscript is a revised version of a previously submitted manuscript. The authors have satisfactorily addressed all concerns of this reviewer. It also appears that issues raised by other reviewers have been addressed appropriately.

One small comment is that in supplemental figure 4, there appears to be a stray star over residue 120. If this is intentional, its meaning is not clear.

We thank the reviewer for the positive comments and pointing us to the stray star on residue 120. The original purpose of the stars was to delineate the F1 and F2 fragments of the RSV F protein. However, the main aim of the figure is to depict the amino acid variations at each position in the entire RSV F sequence so we have removed the F1 and F2 fragment designation from the revised Supplementary Figure 5, as this information was not adding any value.

Reviewer #2 (Remarks to the Author):

The authors have addressed all raised comments adequately.

Reviewer #3 (Remarks to the Author):

The authors have adequately addressed my concerns and the manuscript is much improved, as is the quality of the structure. Please do add units (angstroms, degrees, etc.) to the crystallographic table though.

We have added the units to the crystallographic table and the table was moved to the main manuscript as per the journal requirement and formatting (Table 2).